METHODS AND RESOURCES

# Molecular characterization of nervous system organization in the hemichordate acorn worm *Saccoglossus kowalevskii*

José M. Andrade López[1], Ariel M. Pani[2], Mike Wu[3], John Gerhart[3], Christopher J. Lowe[1]*

**1** Department of Biology, Stanford University, Stanford, California, United States of America, **2** Departments of Biology and Cell Biology, University of Virginia, Charlottesville, Virginia, Unites States of America, **3** Department of Molecular and Cell Biology, University of California, Berkeley, California, Unites States of America

* clowe@stanford.edu

**Data Availability Statement:** All relevant data are within the paper and its Supporting Information files.

## Abstract

Hemichordates are an important group for investigating the evolution of bilaterian nervous systems. As the closest chordate outgroup with a bilaterally symmetric adult body plan, hemichordates are particularly informative for exploring the origins of chordates. Despite the importance of hemichordate neuroanatomy for testing hypotheses on deuterostome and chordate evolution, adult hemichordate nervous systems have not been comprehensively described using molecular techniques, and classic histological descriptions disagree on basic aspects of nervous system organization. A molecular description of hemichordate nervous system organization is important for both anatomical comparisons across phyla and for attempts to understand how conserved gene regulatory programs for ectodermal patterning relate to morphological evolution in deep time. Here, we describe the basic organization of the adult hemichordate *Saccoglossus kowalevskii* nervous system using immunofluorescence, in situ hybridization, and transgenic reporters to visualize neurons, neuropil, and key neuronal cell types. Consistent with previous descriptions, we found the *S. kowalevskii* nervous system consists of a pervasive nerve plexus concentrated in the anterior, along with nerve cords on both the dorsal and ventral side. Neuronal cell types exhibited clear anteroposterior and dorsoventral regionalization in multiple areas of the body. We observed spatially demarcated expression patterns for many genes involved in synthesis or transport of neurotransmitters and neuropeptides but did not observe clear distinctions between putatively centralized and decentralized portions of the nervous system. The plexus shows regionalized structure and is consistent with the proboscis base as a major site for information processing rather than the dorsal nerve cord. In the trunk, there is a clear division of cell types between the dorsal and ventral cords, suggesting differences in function. The absence of neural processes crossing the basement membrane into muscle and extensive axonal varicosities suggest that volume transmission may play an important role in neural function. These data now facilitate more informed neural comparisons between hemichordates and other groups, contributing to broader debates on the origins and evolution of bilaterian nervous systems.

**Funding:** This work was supported by the National Science Foundation (1656628 to CJL). The funders had no role in study design, data collection and analysis, decision to publish, or preparation of the manuscript.

**Competing interests:** The authors have declared that no competing interests exist.

**Abbreviations:** ACR, anterior collar ring; AP, anterior–posterior; BBBA, benzyl benzoate:benzyl alcohol; CalC, calcitonin; CCK, cholecystokinin; CNS, central nervous system; CRH, corticotropin-releasing hormone; DA, dopaminergic; DAT, dopamine transporter; DV, dorsal–ventral; elav, embryonic lethal abnormal visual system; GAD, glutamate decarboxylase; GnRH, gonadotropin-releasing hormone; GPC, glutaminyl-peptide cyclotransferase; GPCR, G protein–coupled receptor; GRN, gene regulatory network; HCR, hybridization chain reaction; HDC, histidine decarboxylase; Hh, Hedgehog; HPA, hypothalamic–pituitary axis; PAM, peptidyl glycine α-amidating monooxygenase; PC2, prohormone convertase 2; PFA, paraformaldehyde; PNS, peripheral nervous system; RT, room temperature; TH, tyrosine hydroxylase; TPH, tryptophan hydroxylase; TRH, thyrotropin-releasing hormone; ZLI, zona limitans intrathalamica; 1GS, 1-gill slit; 3GS, 3-gill slits.

# Introduction

The evolution and origins of animal nervous systems have been a topic of debate in comparative literature for over a century, and the origins of the chordate nervous system have been a topic of particular interest [1–4]. With the emergence of molecular genetic data, the origins of brains and the deep ancestry of central nervous systems (CNS) have been a fascination in the field of evolution of development [5–12]. Classical comparative neurobiology has a rich and long history with broad phylogenetic sampling that was instrumental in developing many of the major hypotheses in neurobiology and nervous system evolution [13]. The spectacular innovations in molecular neuroscience have given unprecedented insights into neural function [14]. However, the focus has been largely on biological models with highly centralized nervous systems and strong cephalization, particularly arthropods and vertebrates [15]. There have been fewer comprehensive molecular studies in phyla outside of these clades, particularly in animals with more sedentary lifestyles and less obvious centralization. Yet, for a broader understanding of nervous system evolution in metazoans, contemporary molecular data from these phyla are essential. This is now particularly pressing, as the generation of comparative developmental patterning data of regulatory genes with key roles in CNS specification and patterning of arthropods and vertebrates has been investigated quite broadly phylogenetically and has by far outpaced parallel contemporary characterizations of neural cell type identity, as well as nervous system structure and function in the same representative species. Application of modern molecular tools of neurobiology to a wider range of species representing a broader sampling of neural systems is now essential and not only tests prevailing views on the evolution of the nervous system but also provides novel opportunities to investigate neural circuit evolution.

Hemichordates are a group of marine invertebrates with sedentary adult life histories that have long been considered to be of considerable importance for understanding the early evolution of chordates [16–18]. Hemichordates were originally grouped within chordates due to proposed morphological affinities [19]. A close phylogenetic relationship with echinoderms based on larval morphology, recognized as early as the late 1800s [20], and all molecular studies have robustly grouped hemichordates as the sister group of echinoderms within the deuterostomes [21–24]. Hemichordates are divided into 2 major lineages: Enteropneusta and Pterobranchia. Enteropneusta are free-living worms that mainly live in burrows and feed through filter feeding or particle ingestion, whereas Pterobranchs are small, largely colonial animals that have received less research attention due to their scarcity and small size [18,25].

The earliest descriptions of the nervous system of enteropneusts using classical staining methods began in the late 1800s [1,26–28] and continued sporadically in the early to mid-1900s [29–34]. Only a few studies have used electron microscopy for more detailed structural observations [35–38]. The enteropneust nervous system is largely intraepidermal with a basiepithelial plexus throughout the animal, more prominent anteriorly in the proboscis and collar, and thickest at the base of the proboscis and proboscis stem. There are 2 cords, one ventral and one dorsal. The ventral cord is basiepidermal, extending from the posterior collar to the posterior of the trunk. The dorsal cord extends along the dorsal midline. At its most anterior extent, the cord runs the length of the proboscis as a superficial cord down to the proboscis stem. In the collar, it is internalized into a subepidermal cord, which in some species resembles vertebrate neural tube [1,37,39]. This collar cord emerges at the base of the collar and extends along the length of the trunk as a basiepidermal cord. The 2 cords are connected by a nerve ring in the posterior collar. Classical studies of hemichordate enteropneust nervous systems have come to contrasting conclusions on the most basic organizational principles of their structure and function. Some studies focused on the importance of the 2 cords, dorsal and

ventral, acting as potential integration centers and concluded that hemichordates possess a CNS [1,33,40], whereas others have focused on the broadly distributed epithelial plexus, with dorsal and ventral cords acting as conduction tracts instead of integration centers [29,31,34]. From work in separate species, Bateson and Morgan concluded that the enteropneust collar cord, an intraepidermal cord, was homologous to the dorsal cord of chordates [1,40] because of the striking similarities to the subepidermal hollow cord that in some species forms by a morphological process that strongly resembles vertebrate neurulation. Later studies in the 1930s to 1950s hypothesize that the dorsal and ventral cord were conduction tracts that lack neural cell bodies [31,34]. Bullock believed that the hemichordate nervous system more closely resembled the nerve net of cnidarians [31], whereas Knight-Jones alternatively proposed that the collar cord may be homologous to the neural tube in chordates but had since been secondarily simplified and is now largely a through tract rather than an integrative center [34]. More recent studies based on comparative morphology have added support to the hypothesis of homology between the hemichordate collar cord and the chordate dorsal cord [37,39].

Recent interest in bilaterian neural system evolution has been driven by body patterning comparisons and the remarkable similarities between the development of CNSs of distantly related nervous systems [6,8,41–44]. Most molecular insights into hemichordate neural development have been inferential, using patterning genes rather than genes with roles in neural cell type differentiation [11,45–50]. However, some studies have focused specifically on genes with established roles in neural specification and differentiation, or cross-reactive antibodies for neural epitopes [11,51–54]. These studies have used pan-neural markers, like *elav* and *synaptotagmin*, and enzymes involved in neurotransmitter synthesis to study the hemichordate nervous system. Although limited, these studies have continued to show evidence of a broad plexus but again have come to contrasting conclusions about the nature of the enteropneust nervous system. There is still no comprehensive study of neural subtypes and their distribution in hemichordates, nor a clear picture of the morphology for different cell types or structure of the neural plexus and cords.

Speculation on the organization of the enteropneust nervous system was reignited with the studies on the spatial deployment of gene regulatory networks (GRNs) with conserved roles in patterning the CNS of many model species during the early development of *Saccoglossus kowalevskii*. The GRN for anterior–posterior (AP) CNS patterning that is well conserved between several protostomes groups and chordates is also conserved in hemichordates along the AP axis even though their nervous systems are anatomically very different [11,46,47,49,50,55]. These genes are expressed in circular bands around the ectoderm in hemichordates, rather than expressed in a tight domain associated with either the dorsal or ventral cords, unlike in vertebrates where they are most prominently expressed in the developing brain and nerve cord [49]. Functional studies have demonstrated the key role of these regulatory networks in body plan regionalization, including nervous system regionalization [46,49], yet there has been no explicit test of how this network regulates nervous system specifically. The neural structures specified by this conserved network evolved independently and represent highly divergent structures despite the tight conservation in their gene regulatory networks. Therefore, it remains an open question as to whether there are some elements of neural conservation despite the overt differences in the organizational elements of their respective nervous systems. It also remains a possibility that there are neural cell type homologies under conserved positional regulation between vertebrates and hemichordates. A closer examination of the molecular neurobiology of hemichordates and the potential link to conserved suites of regulatory genes will contribute to the discussion about how molecular genetic data can be used to reconstruct ancestral neural architectures.

Here, we characterize the expression of multiple neural markers using in situ hybridization, and immunohistochemistry to characterize the location and degree of specialization of the nervous system along both the AP and dorsal–ventral (DV) axes, the extent and structure of the neural plexus. We also used mosaic transgenic approaches to determine cellular morphologies and projection patterns of neural subpopulations. These data provide the first comprehensive description, to our knowledge, of an adult hemichordate nervous system using molecular methods, which will facilitate comparisons to other bilaterians.

## Results

### Extent and regionalization of neural markers along the AP and DV axes

***Elav* expression in the *S. kowalevskii* nervous system.** *Embryonic lethal abnormal visual system (elav)* is a conserved RNA binding protein that has been used in many organisms as a putative pan-neuronal marker [51,56,57]. However, expression of *elav* has been reported in other cell types and does not always mark the entire neural complement [58,59], so caution should be used when only using this marker to fully represent neural complements in developing models. We first sought to identify the distribution of *elav*+ cells in juveniles and adult tissue by in situ hybridization, extending the scope of previous studies at earlier developmental stages [11,51]. This gene has previously been shown to exhibit a tight association with the forming cords on the trunk midlines and expressed broadly in the proboscis ectoderm [11,51,54], but with few cells outside of the cords, posterior to the collar.

We first examined *elav* expression in adult *S. kowalevskii* through a comprehensive series of adult sections in both transverse and sagittal planes (Fig 1A). The transverse plane at mid-proboscis shows circumferential *elav* expression in the most basal layer of the epithelium and more marked staining along the dorsal midline (Fig 1B). The cross section at the proboscis base has a thicker band of expression, supporting previous observations of neural condensation in this region [54] (Fig 1C). At the anterior collar, we detected expression of *elav* throughout the pharyngeal epithelium (Fig 1D–1G) and evidence of the anterior collar nerve ring in the ectoderm (Fig 1D). In addition, there is a prominent layer of *elav* expression dorsal to processes in the dorsal cord, representing cell bodies that runs the length of the collar as an internal collar cord (Fig 1D–1I). Expression in the general ectoderm of collar and trunk is sparse, as described previously in *Ptychodera flava* [54], but cells are detected in the epithelium in the posterior collar and trunk (Fig 1H–1J). Sagittal sections show a gradient of *elav* expression in the proboscis increasing toward the base, an anterior collar ring (ACR) and posterior collar ring, and expression in the pharyngeal endoderm (Fig 1K–1M). The middle sagittal section clearly shows the trajectory of the collar cord (Fig 1L). The enteric nervous system is also visible in the endoderm with dispersed *elav*+ cells in the gut (Fig 1M). At late juvenile stage, characterized by the presence of 3-gill slits (3GS), *elav*+ cells are distributed similarly to their distribution in adults, throughout the proboscis ectoderm, with a concentration at the proboscis base, and also along the entire dorsal and ventral cords (Fig 1N). The proboscis dorsal cord is also visible and extends anteriorly from the collar cord (Fig 1O). The similarities between adult and late juvenile suggests that the juvenile is a reasonable approximation for basic organization of adult neural characters.

Given the comparative interest in hemichordate trunk nerve cords and distribution of neurons in the skin, we examined *elav* expression closely in adults using whole mount in situ hybridization to visualize cell populations that may be difficult to observe in tissue sections. In the ventral cord, *elav* is broadly expressed in a wide band of ectoderm. Rows of clustered cells are present in columns, perpendicular to the cord, along ridges of the epithelium (Fig 1P). On the dorsal side, a thin line of *elav*+ cells is present along the length of the proboscis (Fig 1Q),

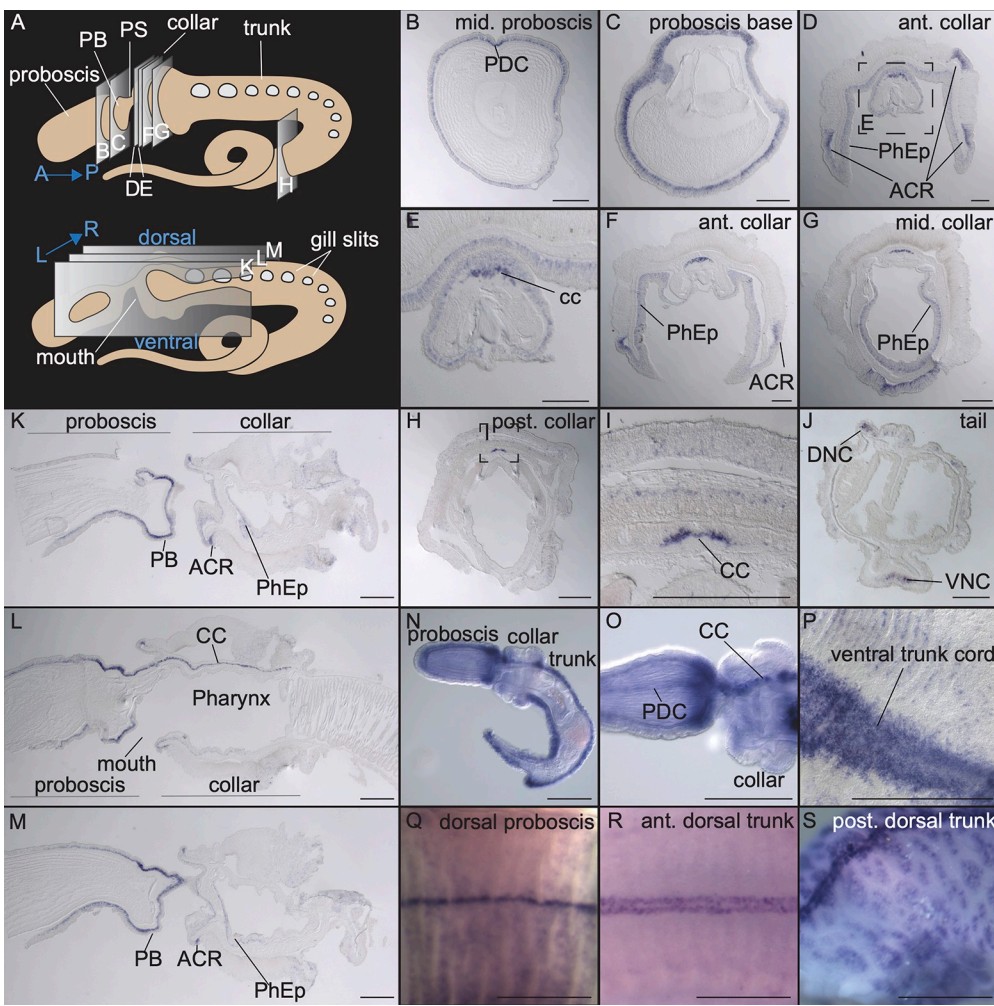

**Fig 1. Pan-neural marker, *elav*, expression. (A)** An illustration representing transverse and sagittal sections taken along the adult body. Dorsal is oriented at the top of the panel unless otherwise specified. **(B, C)** Transverse sections at mid-proboscis (B) and base of the proboscis (C) show circumferential *elav* expression, with most extensive expression at the proboscis base. The proboscis dorsal cord is seen as a small indent at the dorsal side. **(D-I)** Collar sections. **(D-F)** Anterior collar sections with **(E)** higher magnification of the inset from (D) focused on the proboscis stem and start of the collar cord. *Elav* staining in both the ectoderm and pharyngeal epithelium and in the anterior collar ring. **(G-I)** Sections from the mid-posterior collar with **(I)** showing a high magnification of inset from (H). **(J)** Section at the posterior trunk with expression in both dorsal and ventral cords. **(K-M)** Sagittal sections from proboscis to anterior trunk. **(K)** Left of the midline showing the proboscis base and collar revealing the lateral part of the anterior gut and the pharyngeal epithelium (L) is on the midline showing the collar cord, and **(M)** to the right of the midline. **(N, O)** Late juvenile stage, whole mount in situ hybridization. **(N)** Lateral view and **(O)** dorsal view of proboscis and collar. **(P)** *Elav* expression in dissected adult trunk ventral ectoderm showing the ventral cord. **(Q-S)** Whole mount dorsal view of the dorsal cord in different regions of the adult. **(Q)** Adult proboscis **(R)** adult anterior trunk. **(S)** Posterior trunk. All panels show adults except 3GS late juveniles in panels N and O. Scale bars equal 250 μm, except in P-S 500 μm. ACR, anterior collar ring; CC, collar cord; DNC, dorsal nerve cord; PB, proboscis base; PDC, proboscis dorsal cord; PhEp, pharynx endoderm; VNC, ventral nerve cord.

along with the broad expression shown in sections (Fig 1B and 1C). Expression is also detected in 2 narrow rows of cells that flank the dorsal midline along the length of the trunk (Fig 1R and 1S). In the trunk, the *elav+* domain is markedly narrower on the dorsal side than the ventral side. In the posterior trunk, *elav+* cells are also distributed in clusters of cells throughout the epithelium, possibly associated with the calcified granules embedded in this region (Fig 1S).

## Regionalization of neural subtypes

**Expression of neurotransmitter and neuropeptide biosynthesis and transport markers.** While general neural markers like *elav* give a broad picture of the localization of neurons, they do not give any insights into the extent of neural diversity or how neural subpopulations are organized. The expression of genes involved in the biosynthesis and transport of neurotransmitters and neuropeptides provides information about the extent of neural differentiation and the organization of their expression domains along organizational axes. We performed in situ hybridization for components involved in neurotransmitter and neuropeptide signaling, including proteins involved in their synthesis and transport in juvenile, both at the 1-gill slit (1GS) and 3GS stage, and, in some cases, in adult tissue.

Neurotransmitters are small molecules used by neurons as chemical messengers to communicate across cells. They can transmit an excitatory or inhibitory signal synaptically or modulate neuronal activity nonsynaptically using biogenic amines (or monoamines) and amino acids. Genes involved in neurotransmitter synthesis and transport are often highly regionalized in the nervous systems of model bilaterian species.

**Monoamine neurotransmitters.** In *Saccoglossus* juveniles, the location of catecholaminergic neurons is revealed by the expression of *tyrosine hydroxylase* (*TH*) in scattered cells at several different domains in the ectoderm. Early developmental stages were examined previously [51]. In early juveniles, *TH* expression is restricted to isolated cells in the anterior proboscis ectoderm shortly after hatching, and broadly in the anterior trunk, wrapping around the forming first gill slit [49,51]. This pattern continues in the later juvenile stages before hatching with the addition of a thin circumferential line close to the base of the proboscis (Fig 2A). The expression around the gill slits is far more diffuse and extensive than the more cell type–specific staining in the proboscis and collar (Fig 2A, 2A', and 2C). In post hatching juveniles, an additional prominent band of cells forms at the anterior tip of the collar (Fig 2A'). The same general expression domains persist through to adults, but with an expanded anterior proboscis domain (Fig 2B), and additional isolated cells in the collar, with a strong anterior collar domain, and a row of cells at the base of the proboscis (Fig 2B'). To specifically identify dopaminergic neurons, we examined the expression of *dopamine transporter* (*DAT*). In juveniles, *DAT* is coexpressed in many of the same cells in the juveniles (Fig 2C), in the anterior proboscis (Fig 2C'), and in anterior collar (Fig 2C"). *TH*+ cells at the base of the proboscis were conspicuously missing *DAT* coexpression (2C"), raising the possibility they may represent other types of catecholaminergic neurons. However, in older juveniles, *DAT* is expressed in this region and supports the DA neural identity (Fig 2C''').

Two other monoamines are serotonin and histamine. The distribution of *tryptophan hydroxylase* (*TPH*), marking serotonergic neurons, was previously reported in earlier developmental stages [51]. Here, we confirm isolated cellular expression in a broad circumferential domain in the proboscis ectoderm shortly after hatching (Fig 2D). Histamine is synthesized from histidine by *histidine decarboxylase* (*HDC*) and is a marker for histaminergic neurons [60]. In juveniles, expression of *HDC* is sharply defined in the posterior proboscis in a broad ectodermal domain rather than punctate individual cells, which may indicate a broader ectodermal distribution of histamine rather than specifically neuronal (Fig 2E). Additionally, more punctate staining is detected in the ventral ectoderm narrowing to the ventral cord at later stages, and scattered large cells along the dorsal cord in the trunk, which refine to a more posterior territory at later stages (Fig 2E and 2E').

**Amino acid neurotransmitters.** Two major amino acid neurotransmitters are glutamate and GABA. Glutamate is a major excitatory neurotransmitter across bilaterians and non-bilaterians [61–65]. GABA has a conserved role in bilaterians as an inhibitory neurotransmitter in

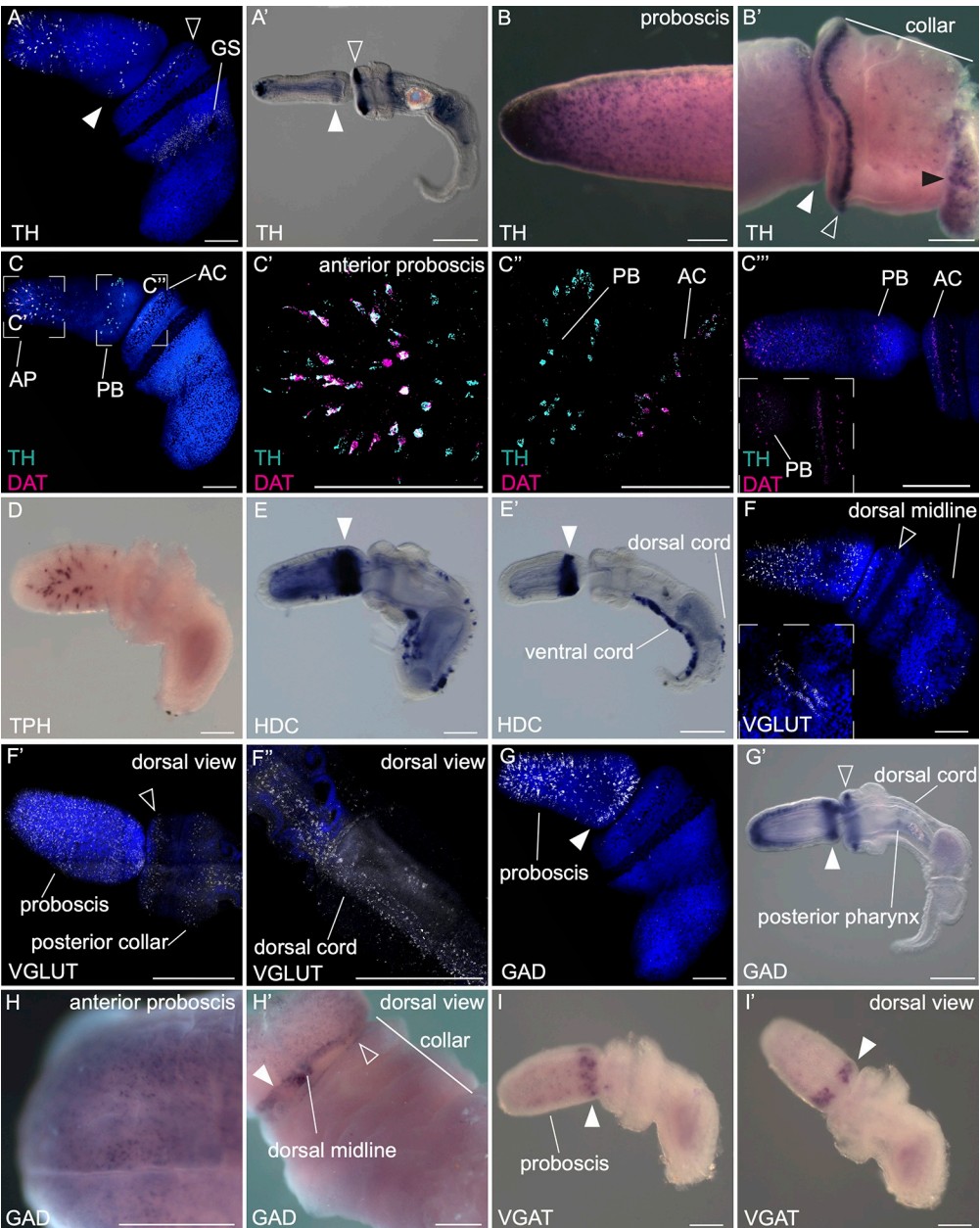

**Fig 2. Gene expression in juveniles and adults for components of neurotransmitter synthesis and transport genes.**
Whole mount colorimetric and fluorescent HCR in situ hybridizations of juvenile and adult samples. (**A-B'**) *Tyrosine hydroxylase*. White arrowheads indicate expression at the base of the proboscis, open arrowhead at the tip of the collar, and black arrowhead at the anterior trunk. (**A**) Expression in early juvenile lateral view, (**A'**) late juvenile lateral view. (**B**) Whole mount expression in adult proboscis, (**B'**) lateral view of adult collar and base of proboscis. (**C-C'''**) Coexpression of *tyrosine hydroxylase* and *dopamine transporter* in (**C**), early juvenile lateral view with (**C'**) a high magnification of the anterior proboscis, and (**C''**), a high magnification of the base of the proboscis. (**C'''**) Proboscis and collar of late juvenile, with inset showing a higher magnificationg. (**D**) Lateral view of an early juvenile showing expression of *tryptophan hydroxylase*. (**E, E'**) Expression of *histidine decarboxylase* (white arrow indicates base of proboscis) in early (**E**) and late (**E'**) juvenile. (**F-F''**) Expression of *vesicular glutamate transporter* in early (F) juvenile, lateral view with lower inset showing dorsal view of the collar and anterior trunk, (**F'**) dorsal view of later juvenile of the proboscis and collar, and (**F''**) dorsal view of the trunk of a late juvenile. (**G-H'**) Expression of *glutamate decarboxylase* in early (**G**) and late (**G'**) juvenile in lateral view, (**H**) whole mount of adult tip of the proboscis showing scattered cell bodies throughout the tip, (**H'**) dorsal view of the adult collar and posterior proboscis. (**I**) Lateral and (**I'**) dorsal view of *vesicular GABA transporter* expression in early juvenile. Closed arrows point to expression at the proboscis base and open arrows point to the anterior collar. Gene expression using HCR are shown as fluorescent

images counterstained with DAPI and all other in situs are shown as a chromogenic staining. Scale bars are 100 μm in early juveniles (A, C-C", D, E, F, G, I-I'), 200 μm in late juveniles (A', C", E', F'-F", G'), and 500 μm in adults (B-B', H-H'). AC, anterior collar; AP, anterior proboscis; HCR, hybridization chain reaction; PB, proboscis base.

both the CNS and peripheral nervous system (PNS) in invertebrates and vertebrates [65–71]. Expression of *glutamate decarboxylase* (*GAD*) has been used to characterize the distribution of GABA–producing neurons [72] along with the GABA transporter *VGAT* [73], whereas glutamatergic neurons have been identified by the expression of its transporter, *VGLUT*.

In *Saccoglossus* juveniles, *VGLUT* is detected in isolated cells in the ectoderm of the proboscis and anterior collar (Fig 2F). *VGLUT* is also detected in 2 rows of cells on either side of the dorsal midline, posterior to the collar at the 1GS stage, and in scattered cells broadly in the posterior trunk. In 3GS juveniles, expression in the proboscis remains broadly dispersed, and there are 2 circumferential lines of cells in both the anterior and posterior collar (Fig 2F'). Expression extends down the trunk, again along the dorsal midline in a broad territory, wider than the extent of the cord defined by the expression of *elav* (Fig 1N, 1O, and 1R), but also dispersed in the general ectoderm (Fig 2F"). *GAD* is expressed in scattered ectodermal cells in defined domains throughout different developmental stages. In early juveniles, *GAD* is expressed throughout the entire proboscis, but most prominently in the anterior tip and at the proboscis base (Fig 2G). At later juvenile stages, the anterior and posterior ectodermal, circumferential domains become more prominent, and a sharp, narrow band of expression at the very anterior lip of the collar is detected in a circumferential domain. At these later stages, expression is now detected in isolated cells in the posterior region of the pharynx and in scattered cells in the dorsal cord of the trunk (Fig 2G'). The juvenile expression is consistent with the patterns found in adult animals: The whole mount in situ hybridization of adult *GAD* expression reveals a dense anterior expression domain of scattered cells that becomes increasingly diffuse down toward the proboscis base (Fig 2H). There is a pronounced ring of cells at the proboscis base that is contiguous except dorsally where the ring terminates with a pair of prominent cell clusters on either side of the dorsal midline (Fig 2H'). The anterior lip of the collar has a similar ring in isolated cells (Fig 2H'). The distribution of the GABA transporter (*VGAT*) in early juveniles shows very similar expression domains to *GAD* at early juvenile stages, with strong expression at the base of the proboscis and cells scattered throughout the ectoderm (Fig 2I and 2I'). However, we do not observe expression in the anterior proboscis where *GAD* is localized at either early or late juvenile stage.

**Neuropeptides.** Neuropeptides are the largest and most diverse signaling molecules ranging from 3 to 40+ amino acids that are involved in neurotransmission, neuromodulation, or hormonal functions [74,75]. Most neuropeptides signal through G protein–coupled receptors (GPCRs) to modulate downstream activities [76,77]. Previous studies have identified an array of conserved neuropeptides and their GPCRs in *S. kowalevskii* [77–81]. Three general neuropeptide synthesis enzymes are *prohormone convertase 2* (*PC2*), *glutaminyl-peptide cyclotransferase* (*GPC*), and *peptidyl glycine α-amidating monooxygenase* (*PAM*), which catalyzes the posttranslational modification of the N-terminal glutamine (*GPC*) or the C-terminal glycine (*PAM*) of peptide hormones [82–86]. Expression of these markers by in situ hybridization reveals the general regional expression of the diverse array of neuropeptides in *S. kowalevskii*. The expression of these enzymes exhibits generally overlapping localization in both early and late juveniles (Fig 3A and 3B), with many cells coexpressing multiple neuropeptide synthesis markers (Fig 3C–3C"'). In early juveniles, *PC2* is broadly expressed in the proboscis, but strongest at the base (Fig 3D). It has a tight ring of expression in the anterior collar, in the collar/dorsal cord, and in the developing ventral cord (Fig 3D). Expression in the later juvenile is strongest at the base of the proboscis, anterior collar ring, and anterior ventral cord, whereas

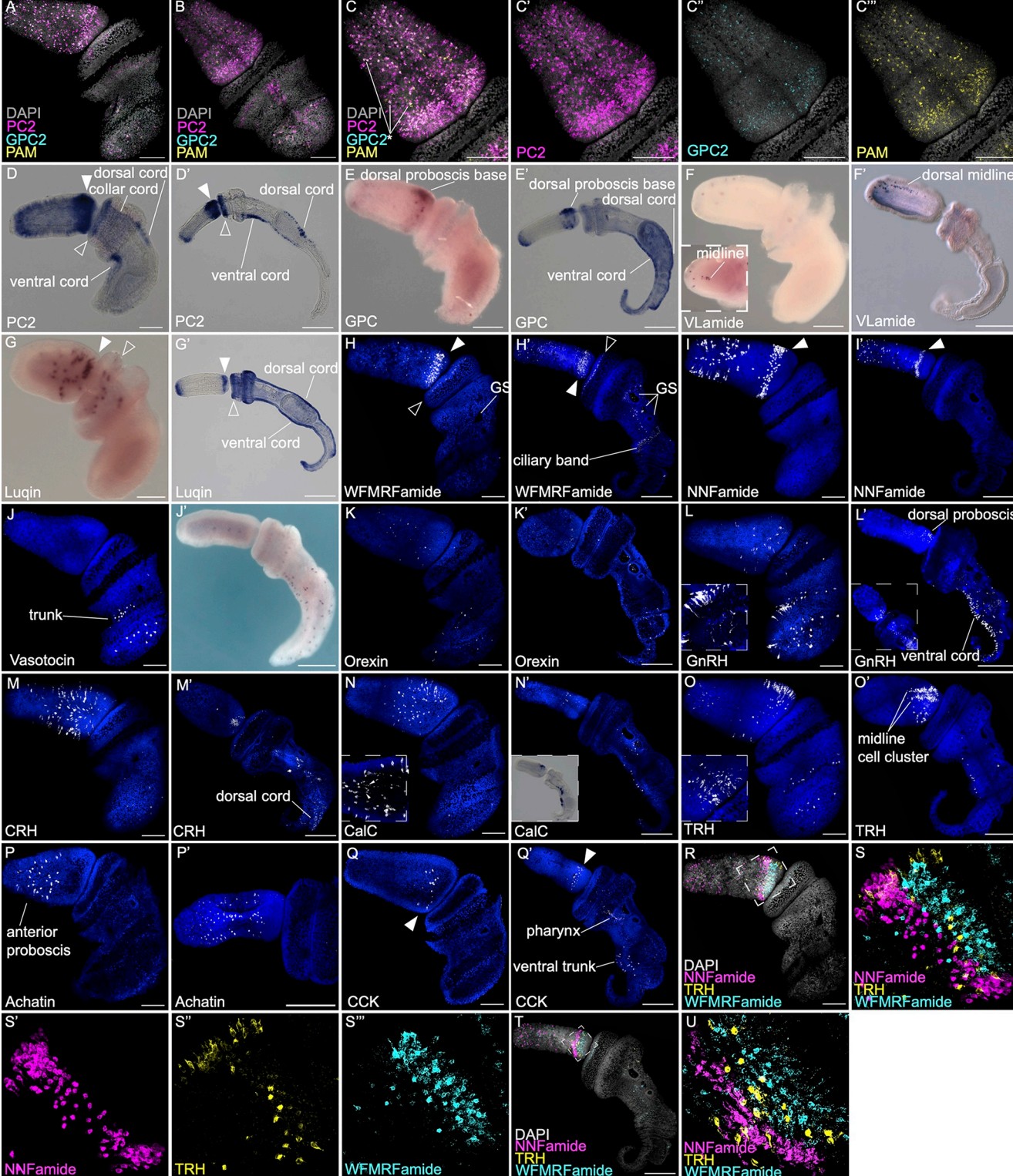

**Fig 3. Gene expression in early and late juveniles for components of neuropeptide signaling including synthesis and transport genes.** (A, B) Coexpression of 3 neuropeptide synthesis genes from early juveniles shown in lateral (A) and dorsal (B) views. (C-C''') Dorsal view of the proboscis showing merged and individual channels from (B). (D-D') *PC2* expression in early (D) and late (D') juvenile. (E-E') Expression of *GPC* in early (E) and late (E') juvenile. (F-F') Expression of VLamide (VIG) in early (F) and late (F') juvenile, and with inset showing dorsal (F) and lateral (F') view. (G-G') Expression of Luqin in early (G) and late (G') juvenile. (H-H') Expression of neuropeptide WFMRFamide in early (H) and late (H') juvenile. (I-I') Expression of the

neuropeptide NNFamide in early (I) and late (I') juvenile, both in lateral view. (J-J') Vasotocin expression in early (J) and late (J') juvenile. (K-K') *Orexin* expression in early (K) and late (K') juvenile, both lateral views. (L-L') GnRH expression in early (L) and late (L') juvenile. Inset in (L) shows proximal axonal projections from GnRH+ neurons in the ventral trunk. Inset in (L') shows a dorsal view. (M-M') CRH expression in early (M) and late (M') juvenile, inset shows expression at the heart-kidney complex at the proboscis base. (N-N') CalC expression in early (N), with inset showing a lateral view of the proboscis base with proximal axon projections, and late (N') juvenile, with inset showing expression as a chromogenic staining. (O-O') TRH expression in early (O), with the dorsal proboscis base view in the inset, and late (O') juvenile. (P-P') Achatin expression in early (P) and late (P') juvenile, both lateral views, with only the proboscis and collar shown in (P'). (Q-Q') CCK expression in early (Q) and late (Q') juvenile, lateral views. (R-U) Coexpression of the neuropeptides NNFamide, TRH, and WFMRFamide in early (R-S''') and late (T-U) juvenile, with insets (S, U) showing merged maximum intensity projection of demarcated expression patterns and individual channel (S'-S''') for the early juvenile. Closed arrows point to expression at the proboscis base and open arrows point to the anterior collar. Gene expression using HCR are shown as fluorescent images counterstained with DAPI in blue, and all other in situs are chromogenic. Scale bars are 100 μm in early juveniles (A-D, E, F, G, H, I, J, K, L, M, N, O, P, Q, R) and 200 μm in late juveniles (D', E',F', G', H', I', J', K', L', M', N', O', P', Q', T). CalC, calcitonin; CCK, cholecystokinin; CRH, corticotropin-releasing hormone; GnRH, gonadotropin-releasing hormone; GPC, glutaminyl-peptide cyclohydrolase; HCR, hybridization chain reaction; PC2, proprotein convertase 2; TRH, thyrotropin-releasing hormone.

expression in the anterior proboscis and dorsal cord becomes less prominent (Fig 3D'). *GPC* expression is more dorsally localized at the proboscis base with more punctate expression in the ventral ectoderm (Fig 3E and 3E'), whereas *PC2* expression is more uniform in a thick band around at the proboscis base (Fig 3D and 3D'). We also observed expression in the more posterior regions of the dorsal and ventral cords. Together, these enzymes are expressed at the 5 major regions of the ectoderm: anterior proboscis, posterior proboscis, anterior collar, dorsal cord, and the ventral cord and overlap in their expression domains in the proboscis and collar with several of the neurotransmitter systems.

*S. kowalevskii* contains many neuropeptides [77–81] with a signature C-terminal sequence including VIamide, Luqin (RWamide), WFMRFamide, and NNFamide. We infer the localization of neuropeptides by the localization of mRNA for the precursor proteins or propeptides. VIamide is characterized by an anterior, dorsal domain of expression in early and late juveniles, with isolated cells along the dorsal midline from the anterior tip to about half way down the proboscis (Fig 3F and 3F'). The conserved bilaterian neuropeptide luqin (Luq), subsequently lost in the chordate lineage [79], belongs to the FMRFamide and RFamide-like neuropeptide family that was first discovered in mollusks [87,88]. Comparative studies across bilaterians suggest a shared role in chemosensory and locomotion control through flask shaped, ciliated RFamide neurons [89]. Luq is initially expressed in the posterior proboscis and anterior collar at the 1GS stage, but expression is later detected at the anterior and posterior proboscis, anterior collar, the dorsal cord, and the ventral cord extends toward the post anal tail in 3GS juveniles (Fig 3G and 3G'). WFMRFamide expression appears as scattered cells in the anterior proboscis and strong circular bands at the proboscis base and anterior collar in both early and late juveniles (Fig 3H and 3H'). Expression also appears in the trunk around the gill slits and ciliary band in late juveniles (Fig 3H'). NNFamide is expressed in the proboscis ectoderm in 2 main domains; the most prominent is in a strong, circumferential, horseshoe-shaped band with expression absent in the most dorsal territory, close to the base of the proboscis (Fig 3I and 3I'). The second domain is defined by scattered individual cells visible in the entire anterior half of the proboscis ectoderm, all the way to the anterior tip.

The hypothalamic–pituitary axis (HPA) is a major neuroendocrine system associated with the regulation of many biological and physiological mechanisms including regulating metabolism, immune system, stress, reproduction, growth, and development by acting on endocrine glands like the adrenal, gonads, and thyroid [90–93]. Neurons from the hypothalamus regulate the pituitary and downstream organs including the gonads, adrenal gland, and thalamus by stimulating the release of neuropeptides conserved across bilaterians [77,79,81,94,95]. The *S. kowalevskii* genome contains many of these conserved neuropeptides including vasotocin, orexin, gonadotropin-releasing hormone (GnRH), corticotropin-releasing hormone (CRH), thyrotropin-releasing hormone (TRH), and calcitonin (CalC), which have all been identified

in previous studies [77,81,96,97]. Vasotocin and orexin are 2 bilaterian-conserved neuropeptides secreted in the hypothalamus in vertebrates. Vasotocin is expressed in scattered cells at the trunk in early and late juveniles (Fig 3J and 3J'), whereas orexin is broadly expressed throughout the epithelium in early and late juveniles, with minimal expression around the collar (Fig 3K and 3K'). GnRH stimulates secretion of gonadotropins from pituitary neurons to regulate gametogenesis and gonadal development in vertebrates [98,99]. In *S. kowalevskii*, GnRH+ neurons are located in scattered cells at the proboscis base and in a similar domain to vasotocin at the trunk in early juveniles (Fig 3L). Neurons in the trunk have a visible proximal projection into the plexus, shown in the in the inset image (Fig 3L). Late juveniles have a strong ventral cord expression, with few cells along the dorsal cord and dorsal proboscis base (Fig 3L'). CRH neurons in the vertebrate hypothalamus signal to the anterior pituitary and stimulating the release of adrenocorticotropic hormone into the bloodstream to regulate the stress response [100,101]. CRH is expressed throughout the proboscis in early juveniles (Fig 3M) and becomes more restricted to the posterior dorsal proboscis in late juveniles (Fig 3M'). CalC helps control plasma calcium levels to regulate bone remodeling and metabolism in vertebrates [93,102,103] and is thought to have an ancestral role in regulating biomineralization [104]. CalC expression is seen in the posterior part of the proboscis in scattered cells in both early and late juveniles (Fig 3N and 3N') and along the ventral cord in late juveniles (Fig 3N'). White arrows in the inset panel in Fig 3N show posterior projections of these neurons. TRH is expressed in scattered cells in the anterior proboscis, strong expression in the dorsal proboscis base, and along the trunk in early juveniles (Fig 3O). Cells at the trunk become restricted to the ventral cord in late juveniles (Fig 3O'). In both early and late juveniles, there are 2 dominant cell clusters adjacent to the dorsal proboscis base.

Other conserved neuropeptides within bilaterians include achatin and cholecystokinin [77,79,95]. Achatin shows restricted expression in large isolated cells distributed in the anterior proboscis (Fig 3P and 3P'). Cholecystokinin (CCK) is a gastrointestinal hormone peptide that has an ancient role in regulating feeding [105]. CCK is expressed in the proboscis base in early juveniles (Fig 3Q) and later has broader expression in the pharynx endoderm and ventral trunk ectoderm in late juveniles (Fig 3Q').

Because many of the neuropeptides show dense expression at the base of the proboscis, we tested whether there was coexpression of multiple peptides or if each neural subtype was associated with specific neuro peptides. We performed multiplexed hybridization chain reaction (HCR) for 3 peptides, NNFamide, TRH, and WFMRFamide (Fig 3R–3U), all expressed in similar domains. The anterior proboscis shows nonoverlapping expression of the 3 markers evenly distributed across the epithelium across both stages (Fig 3R and 3T), and the posterior proboscis is composed of nonoverlapping rings of expression of individual neuropeptides (Fig 3S and 3U). The minimal neuropeptide coexpression suggests that neuropeptides may be good markers for specific neural cell types in the proboscis.

Unlike neurotransmitters, which preferentially exhibit anterior expression, neuropeptides show more extensive posterior expression along the ventral and dorsal cords. Expression data for neurotransmitter and neuropeptide synthesis and transport markers suggest that *S. kowalevskii* has a strongly regionalized nervous system, with an increased neural cell type diversity in 5 main territories, the anterior proboscis, posterior proboscis, anterior collar, dorsal cord, ventral cord, but most prominently, the dorsal proboscis base.

## Plexus and neural cord organization

In the previous section, we described the expression of many important genes involved in neural function by in situ hybridization. These data provide useful information about

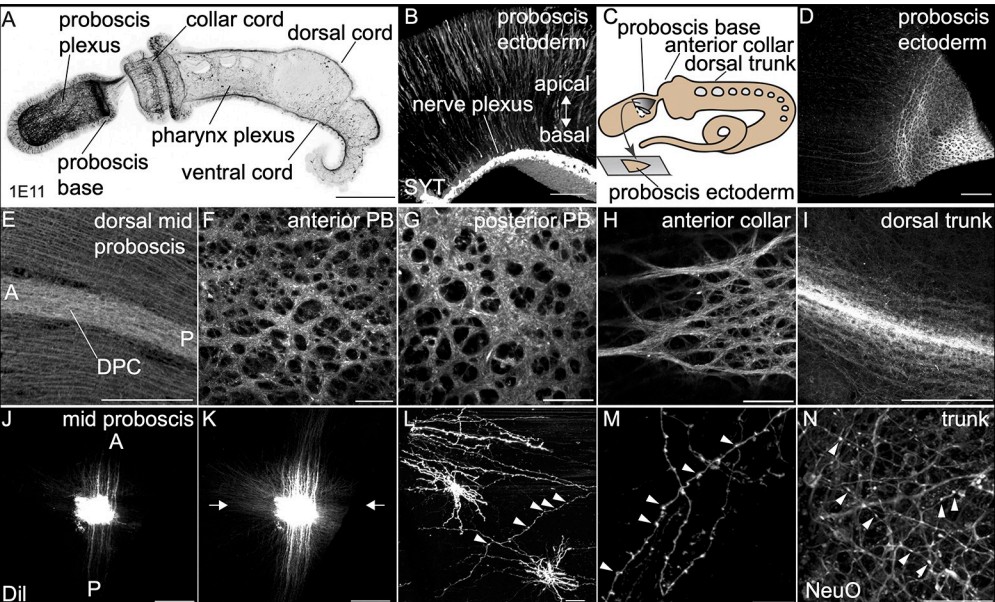

**Fig 4. Neural plexus organization.** Visualization of the neural plexus in fixed and live tissue using an anti-synaptotagmin (1E11) antibody. (A-I), DiI (J-M), and NeuO (N). Juveniles are imaged in whole mount, and adult ectoderm by flat mount following dissection and imaging from the basal surface. (A) Maximum intensity projection of a late juvenile stained for 1E11, lateral view. (B) The adult ectoderm at the base of the proboscis. (C) Illustration showing the performed excision of the posterior proboscis ectoderm from an adult animal. (D-I) Expression of synaptotagmin in dissected adult ectoderm in different regions of the body: (D) posterior proboscis base dissected as shown in C, (E) dorsal proboscis showing dorsal proboscis cord (DPC), (F) anterior region of the proboscis base, (G) posterior region of the proboscis base, (H) the anterior collar, and (I) the dorsal trunk. (J-M) Lipophilic dye (DiI) injections into adult fixed tissue reveals dye diffusion across the adult plexus in the proboscis ectoderm in (J) and (K), and neurite cell morphology in (L) and (M). The white arrows in J and K show the direction of projections, mostly AP, in the most apical apical region of the plexus in J and in all directions in the basal plexus in K. (N) Live neural marker, NeuO, showing similar puncta along neurites seen in fixed tissue in (L) and (M). Scale bars represent 250 μm in A, D, E, I-K and 25 μm in B, F-H, L-N. Arrow heads in L-N point to the puncta along neurites.

specialization and location of neuronal cell bodies but tell us little about neural morphology, neurite and axonal projections, and the general structure of the nervous system. To begin to investigate nervous system structure and function, we performed immunohistochemistry with a monoclonal antibody (1E11) to the pan-neural marker *synaptotagmin*, which was developed from the radial nerve extract of asteroid, *Asterina pectinifera* [106], to visualize cell morphology and neurite projections in juveniles and adults (Fig 4A). Previous comparative work has validated the cross reactivity of this antibody localizing to neurons in other ambulacrarian taxa, broadly in echinoderms and in another species of hemichordate [107,108].

Whole mount immunohistochemistry of late juveniles at the 3GS stage revealed a complex and structured nervous system with a dense basiepithelial nerve plexus throughout the proboscis, but thicker at the base (Fig 4A). The plexus is also very prominent in the collar ectoderm, but less in the trunk. An endodermal plexus is visible throughout the pharynx (Fig 4A). The subepidermal collar cord is clearly visible extending from the thick plexus at the dorsal posterior ectoderm of the proboscis into the superficial dorsal cord, which begins in the posterior collar and runs the length of the trunk terminating at the anus (Fig 4A). Neurites and axons are well labeled, but cell bodies had weak signal (Fig 4B). The wider superficial ventral cord at the posterior collar/anterior trunk boundary extends posteriorly throughout the entire ventral midline of the animal.

We further utilized 1E11 to determine the structure of the nervous system in adults. As many of the neural cell type markers exhibit localized domains of expression in the proboscis, we first investigated 1E11 expression at this site. The ectodermal plexus in this territory was peeled off from the underlying proboscis mesoderm in fixed adults, and whole mount immunohistochemistry was carried out on these tissue fragments (Fig 4C). In the general proboscis ectoderm, we observe a well-organized plexus with parallel bundles of processes running along the AP axis and regularly spaced connectors projecting laterally between the bundles (Fig 4E). This organization is remarkably similar to the drawings of Knight-Jones [34]. The dorsal superficial cord is visible as a thickening of this plexus (Fig 4E). The neurite bundles projecting along the AP axis observed in the mid-proboscis exhibit a striking transition in plexus structure: a complex architecture of thicker neurite bundles forming a more disordered mesh that shows clear organizational differences along the basal/apical axis of the plexus. This can be clearly observed in Fig 4D where the flat mount of the ectoderm shows the parallel projections, on the left of the panel, projecting into the complex plexus at the proboscis base, at the right of the panel. Higher magnification shows the complex architecture of this territory (Fig 4F and 4G). The structure of this domain is very reminiscent of the anterior territory of the plexus structure in the acoel *Hofstenia miamia* [109], and the holes in the plexus may represent extensions from epithelial cells that attach to the basement membrane as reported from early EM studies [35]. Moving posteriorly into the collar, we observed a similar, structured plexus to the proboscis base, but with less densely packed tracts (Fig 4H). A z-stack (S1 Movie) in this region clearly shows the structured nature of this plexus. Further posterior in the trunk, the plexus is less extensive than in the proboscis or collar, with the dorsal and ventral cords being the most prominent features representing condensations of the plexus along both midlines (Fig 4I).

1E11 staining reveals many general aspects of the distribution and organization of the neural plexus, but little resolution of individual neurites because of the densely packed neurite bundles. We used injection of lipophilic dye (DiI) into adult fixed tissue in the mid-proboscis to look more closely at the morphology and directionality of neurite projections. Individual neurites extend from the injection site in all directions, but with prominent neurite bundles projecting along an AP axis (Fig 4J and 4K), matching what was observed with the IE11 antibody at the proboscis (Fig 4D and 4E). The plexus is clearly structured, with the imaging stacks showing different projection patterns according to the position along the apico/basal axis in the plexus (Fig 4J and 4K). Lower volume injections of DiI in the proboscis ectoderm imaged at higher magnification reveal many neurites with regular puncta along their length (Fig 4L and 4M). As these swellings may represent fixation artifacts, we utilized a live neural stain, NeuO, a membrane-permeable fluorescent probe [110]. We observe similarly labeled neurites/axons in the trunk revealing regularly spaced swellings along the neurites/axons (Fig 4N). The similar structures detected along the processes using both fixed (DiI) and live (NeuO) tissue suggest that these structures may represent either en passant synapses [111] or varicosities involved in volume release [112].

## Morphology and projection of neural subtypes

Our in situ hybridization data for neurotransmitter and neuropeptide synthesis and transport markers clearly demonstrate that *S. kowalevskii* has strong regional specialization of its nervous system and also revealed the location of cell bodies characterized by specific neurotransmitters and neuropeptides. However, these data provide no information about the directionality or length of projections, and in situ hybridization data rarely provide information about cellular morphology. We used 2 different methods to investigate details of neural subtype morphology and projections, cross-reactive antisera to neurotransmitters and neural transgenics.

## Cross-reactive antisera

**Serotonergic neurons.** We used an antibody raised to 5-HT to label serotonergic neurons in 3GS stage *S. kowalevskii* (Fig 5A). Previous work using this antibody has already investigated the distribution of the serotonergic nervous system in early developmental stages, in juveniles and limited sections in adults, and our data confirm expression and extend sampling of these studies [51,52,113]. In juveniles, the serotonergic nervous system is composed of primarily receptor, flask-shaped bipolar neurons broadly dispersed in the proboscis ectoderm and collar, but more sparsely in the trunk. In the proboscis, while staining is broadly scattered, there are no cell bodies detected either in the most apical ectoderm or at the base of the proboscis (Fig 5A and 5C). Neurons project into the underlying neural plexus, although it is not possible to trace the full length of individual axons due to large numbers of projections in the plexus. However, proboscis neurons appear to generally project posteriorly. In the collar, cell bodies are organized into 3 rings, two at the anterior and one at the mid-collar (Fig 5A and 5D). These neurons also have a single dendrite and sensory cilium projecting out of the epithelium, and a single axon projecting basally into the neural plexus. There are positional differences in cell body location within the epithelium; some cell bodies are located basally and others more apically (Fig 5B). In the collar and proboscis, we see no dorsoventral differences in the distribution of cell bodies; however, cell bodies in the trunk are positioned on the dorsal side, mostly lateral to the dorsal cord and project ventrally into the ventral cord (Fig 5E). In the anterior part of the trunk, there are also cell bodies scattered in the ectoderm more laterally around the gill slits (Fig 5A). A z-stack of the serotonergic nervous system in this region again clearly shows the structured nature of this plexus, as well as dorsal sensory cells projecting laterally along the trunk toward the ventral cord (S2 Movie).

We further extended our analysis into adult animals. Whole mount immunohistochemistry in adults show a far more extensive serotonergic population than in juveniles but generally confirm a similar expression pattern. Fig 5F shows a view of the mid-proboscis, showing broad distribution of cell bodies projecting into the underlying epithelium, with projections in all directions, but most projecting posteriorly, similar to the organization revealed from 1E11 (Fig 4E). The base of the proboscis (the same specimen as Fig 4D and 4E) double labeled for both 1E11 and 5HT (Fig 5G and 5H) reveals that serotonergic axons form a subset of the complex network of bundles. The morphology of the neurites shows swellings along the length of the processes similar to what we demonstrated with DiI labeling. In the collar (Fig 5I), staining is absent in the anterior lip, but just posterior, there is a ring of expression in the ectoderm, and broad ectodermal labeling throughout the collar ectoderm. Cell bodies are more broadly dispersed throughout the collar epithelium rather than in discrete rings as in 3GS juveniles. Note that these findings contrast with the lack of neural staining from *elav* in this region (Fig 1). In the posterior trunk, the epithelium is ruffled, and we observed patches of cell bodies in the lateral body wall and an extensive, but thin, plexus throughout the trunk epithelium (Fig 5J).

We next investigated the labeling of the serotonergic neurons in representative cross sections along the adult body, counterstained with phalloidin (Fig 5K). In the posterior proboscis, we observe clear dorsoventral asymmetry in plexus thickness, with the dorsal region thicker than the ventral territory, and with cell bodies distributed more ventrally in this plane of section (Fig 5L). In the mid-collar (Fig 5M), the neural plexus is extensive in both the ectoderm and the endoderm but is thinner than the proboscis plexus (Fig 5L). The dorsal cord is clearly visible (Fig 5N): DAPI labels the soma of the cord positioned above the neurites that project through the cord, and as reported elsewhere, there is no obvious cord lumen. We did not observe any 5HT+ cell bodies in the cord soma in the sections we examined. As observed in

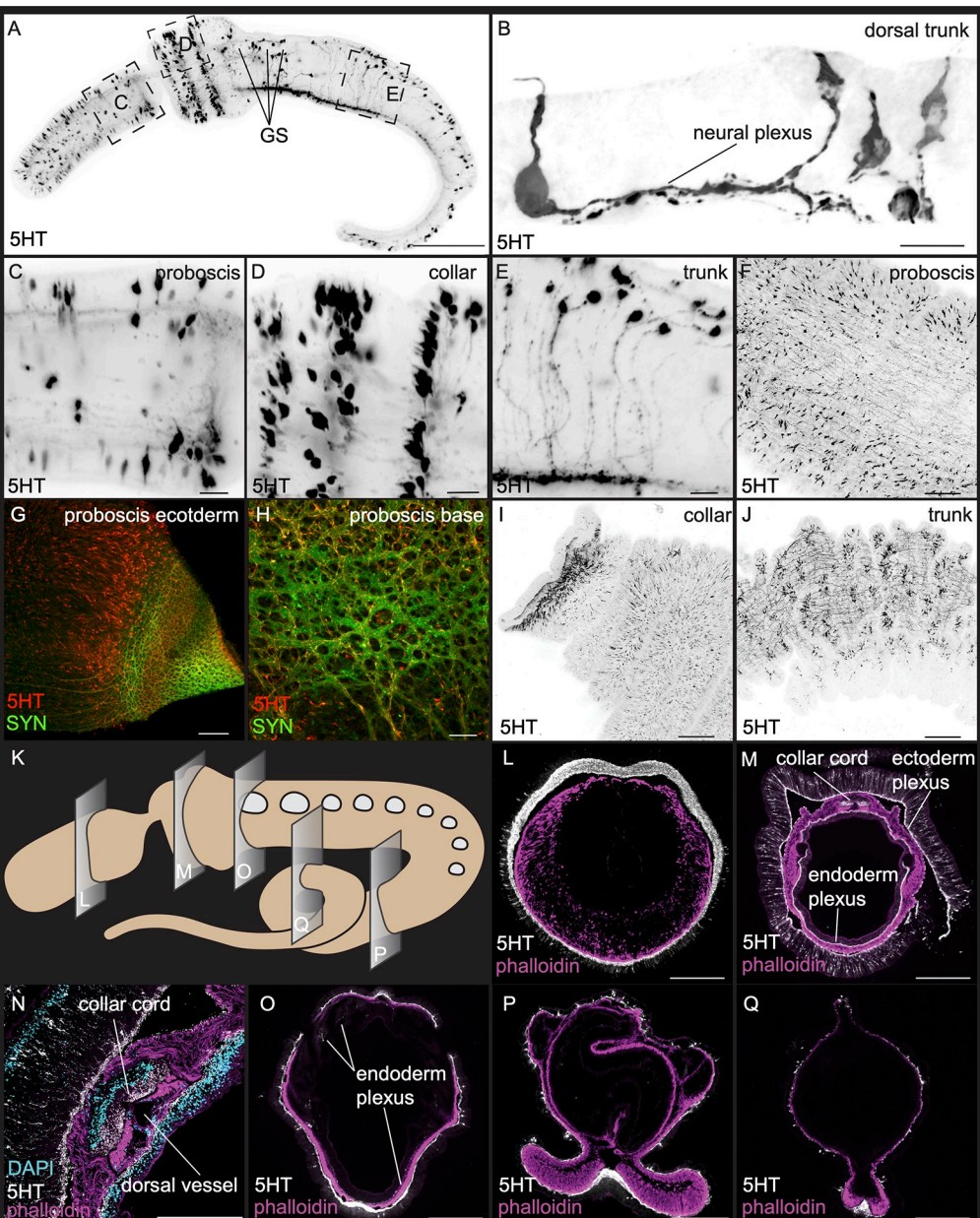

**Fig 5. Serotonergic nervous system.** Investigation of the serotonergic nervous system in late juveniles (A-E) and adults (F-Q) using anti-5HT. (A) Maximum intensity projection of a late juvenile lateral view. (B) Anterior dorsal trunk region just posterior to the collar showing bipolar neurons with a sensory cilium projecting toward the apical region of the epithelium and a single neurite projecting basally into the neural plexus. (C-E) Inset panels showing the (C) posterior proboscis, (D) collar, and (E) trunk. Images are oriented in an anterior (left)-to-posterior (right) and a dorsal (up)-to-ventral (down) direction. (F-J) Adult ectoderm in the proboscis, (F) mid-proboscis, (G, H) base of the proboscis double labeled with *synaptotagmin*, and the same specimen as in Fig 3E. (I) Collar and (J) trunk. (K) An illustration showing the transverse sections carried out in adult tissues. All sections oriented with dorsal at the top of the panel with phalloidin labeled in purple, 5HT in white, and DAPI in blue. (L) Proboscis, (M) mid-collar, (N) dorsal collar cord, (O) anterior trunk, and (P) and (Q) showing the posterior trunk. The anterior to posterior orientation are from left to right in adult tissue in F-J. The proboscis was removed in I to only show the collar and part of the trunk. Scale bars represent 500 μm in F, I, J, L, M, O-Q; 200 μm in A, G, N; and 20 μm in B-E, H.

the whole mount (Fig 5M), there is a broad distribution of cell bodies projecting into the plexus, without any dorsoventral differences. The pharyngeal epithelium also shows a prominent plexus throughout the pharynx, yet only a few isolated cell bodies are associated with the endoderm in these sections. Sections in the anterior trunk show cell bodies sparsely scattered throughout the ectoderm with a thin ectodermal plexus that thickens ventrally and the dorsal cord showing far fewer axons labeled than in the ventral cord. The endodermal plexus is very sparse (Fig 5O). In the posterior trunk, labeling of axons in the ventral cord is more prominent and a thin plexus is detected throughout the epithelium with scattered, isolated cell bodies (Fig 5P and 5Q). No endodermal plexus was detected.

**GABAergic neurons.** We used a GABA polyclonal antibody to stain GABAergic neurons in juveniles, as has been previously demonstrated in another enteropneust species, *P. flava* [54]. To address concerns about antibody binding specificity, we compared GABA antisera reactivity with the in situ hybridization for GAD (Fig 2). We observed good concordance between the antibody and in situ hybridization localizations. The GABAergic nervous system in juveniles is concentrated both in the anterior and posterior proboscis ectoderm (Fig 6A–6B). In the collar, there are 2 ectodermal rings of cells at the lip of the collar (Fig 6A). In the trunk, there are isolated neurons along the dorsal midline in both the ectoderm and endoderm (Fig 6C). These cells appear similar in morphology to 5HT+ cells, bipolar with a single neurite extending from the cell body to the apical region terminating with an apical cilium, and the axon descending into the plexus on the basal side. There are also dorsal endodermal neurons with a neurite projecting to the apical surface of the endoderm but with the cell body embedded in the endodermal plexus (Fig 6C). Further posteriorly, there are prominent axonal projections around the gill slits and into the ventral cord (Fig 6D).

The number of cell bodies at the proboscis, collar, and trunk increases substantially in adults. The base of the proboscis forms a similar pattern of GABA+ neurite bundles to those detected from 1E11 and 5HT (Fig 6E). The adult collar has a more expansive concentration of neurons with the anterior collar ring and scattered cell bodies throughout the collar ectoderm (Fig 6F). GABAergic neurons are labeled in gill bars (Fig 6G) with ventrally projecting axons. To directly image the dorsal cord, we dissected the cord from the ectodermal tissue, keeping the nerve bundles intact. This revealed many neurites projecting in an anterior-to-posterior direction along the length of the cord, with some neurites projecting laterally (Fig 6H).

**FMRFamidergic neurons.** We used the rabbit polyclonal anti-FMRFamide to identify potential FMRFamidergic neurons in 3GS stage embryos. This antibody has been shown to be cross-reactive in a diverse set of bilaterians including echinoderms, where reactivity is observed in the radial nerve cord, tube feet, apical muscle, intestine, and the esophagus nerve plexus [114]. While the exact epitope that is recognized is uncertain, some studies have also found that the FMRFamide antibody exhibits cross-reactivity with SALMFamides and GFSKLYFamide in the sea cucumbers [115,116]. Therefore, the possible affinity to other neuropeptides must be considered in *S. kowalevskii*. The labeling shows significant overlap with the in situ data for *PC2* and *GPC*, the 2 enzymes involved in the processing of neuropeptides, so these data provide information about the projection of a subset of neuropeptides within the complement in *S. kowalevskii*.

The peptidergic labeling from this antibody shows a broad circumferential distribution of neurons in the proboscis but with a concentration at the proboscis base of both cell bodies and axons (Fig 6I). Neurons at the proboscis base have a flask-shaped morphology like the morphology of many other neurons described in this study, with a single cilium extending into the outer ectoderm (Fig 6I). These neurons appear to project posteriorly down the proboscis stem and along the dorsal collar cord, with some projections following the nerve plexus along the collar epithelium (Fig 6I and 6J) connecting with the ventral cord along the trunk (Fig 6I). Cell bodies are detected in the ventral cord with axons running the length of the cord (Fig 6K). In

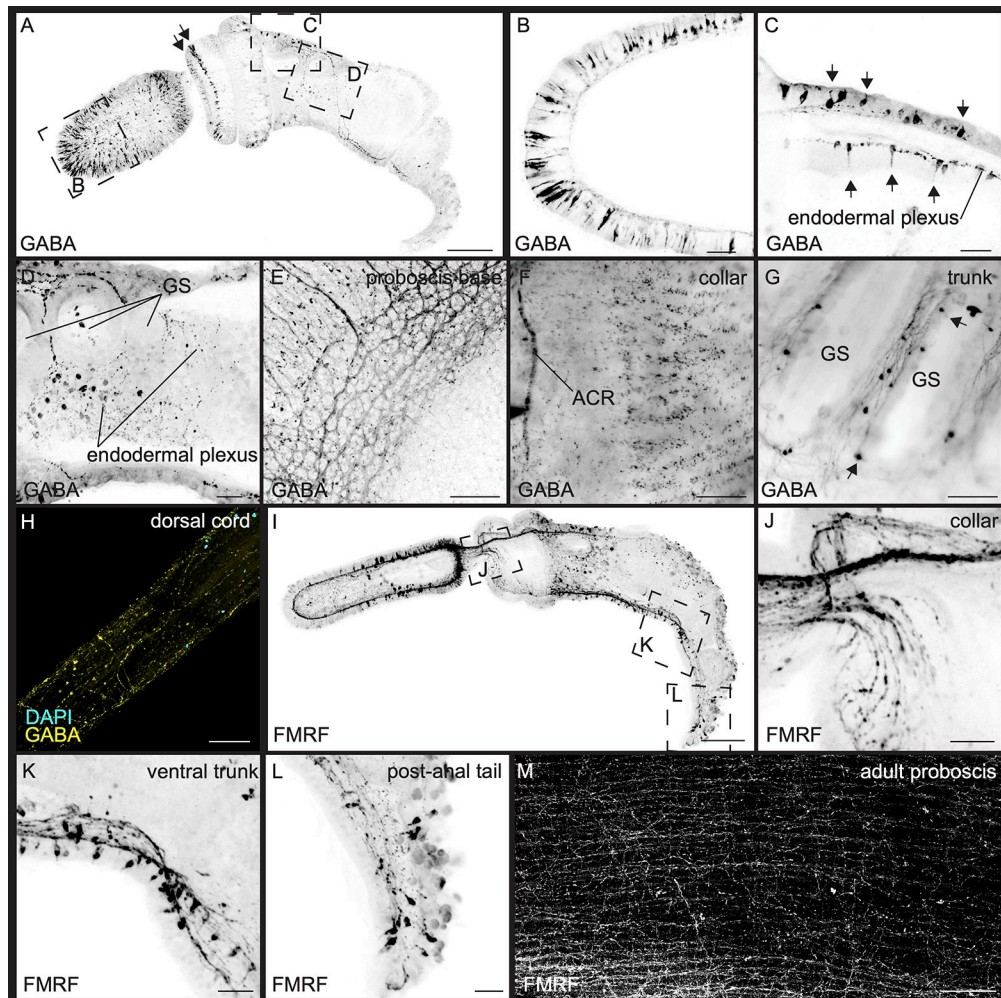

**Fig 6. GABAergic and FMRFamidergic nervous system.** Distribution of GABAergic and peptidergic neurons in juveniles and adults using anti-GABA and anti-FMRFamide polyclonal antibodies. (A-D) Distribution of GABA in juveniles. (A) Maximum intensity projection of a late juvenile (black arrows in A point to the ring of GABA+ neurons at the anterior collar), and inset panels from (A) along different regions: (B) anterior proboscis, (D) anterior dorsal trunk (arrows indicating the dendrites projecting to the outer epithelium of both the ectoderm and endoderm), and (D) endodermal plexus in the pharyngeal gut of the anterior trunk, with neurites projecting around the gill slits (GS). (E-H) Dissected adult tissue at (E) the proboscis base, (F) collar, (G) gill slits at the trunk, and (H) dissected dorsal collar cord. (I) Maximum intensity projection of a late juvenile labeled with anti-FMRF-amide; panels show the position of the subsequent high magnification images, (J) anterior collar and proboscis stem, (K) ventral trunk/ventral cord, and (L) postanal tail. (M) Adult proboscis plexus. Anterior to posterior orientation is from left to right in E-G and M. Scale bars are 100 μm, except 20 μm in panels B-D and J-L.

the postanal tail of the juvenile, axons are clearly projecting anteriorly (Fig 6L). We examined labeling of the adult nervous system by dissection of the ectoderm from the mesoderm in the proboscis and imaged the plexus (Fig 6M). This revealed an extensive plexus throughout the proboscis ectoderm with a general trend of bundled projections along the AP axis but with many lateral neurites connecting these bundles projecting both long and short distances.

### Neural transgenes

**Synapsin transgene.** To visualize neurons in higher resolution, we generated a Synapsin construct to drive expression of eGFP. Synapsin is a synaptic vesicle transmembrane protein

and a marker of differentiated neurons [117,118]. Our construct was designed using 8 kilobases (kb) upstream of the start site of the synapsin-2-like gene (XP_006820290.1) [119]. We examined transgene expression in over 50 $F_0$ juveniles at a range of developmental stages. The transgenic animals exhibit mosaic incorporation of the transgene and show neuronal staining prominently throughout the proboscis ectoderm, sometimes in the dorsal region of the trunk and collar (Fig 7A and 7B). Because of the mosaicism, transgenic animals ranged from a single labeled neuron to hundreds of labeled cells. From these transgenics animals, we identified the range of neural morphologies and the length and directionality of their neural processes. The most common neural morphology we observed with this transgene were large bipolar sensory neurons, like those observed with 5HT immunohistochemistry. We found these neurons throughout the ectoderm but at the highest density in the proboscis. In a separate synapsin: eGFP juveniles, we detect large bipolar neurons in the anterior tip of the proboscis projecting posteriorly toward the proboscis base (Fig 7B). The anterior proboscis contains a neural plexus with axonal swellings (white arrows, Fig 7C). At the dorsal proboscis base, neurons have a flask-shaped morphology, with a rounded nucleus close to the neural plexus and with either a single axon (white arrow in inset Fig 7D) or an axon that splits into an anteriorly and posteriorly projecting extension (gray arrow in inset Fig 7D). We also identified a group of elongated sensory neurons in the posterior collar with a single axon projecting anteriorly into the cord (white arrows, Fig 7E). The projections of this group of neurons are likely involved in relaying posterior sensory information from the trunk to the anterior part of the animal with most axonal termini at the base of the proboscis. We rarely detected unipolar neurons, but one is shown associated with the plexus projecting anteriorly into the dorsal cord toward the proboscis (gray arrow, Fig 7E). Elongated bipolar neurons are detected in the far posterior ectoderm and project anteriorly along the dorsal midline, likely along the dorsal cord (Fig 7F).

Many of the bipolar neurons have prominent swellings along the axons projecting into the basiepithelial plexus (Fig 7G) as was reported for DiI and serotonin. Many of these varicosities are located close to the basement membrane of the plexus, so it is possible that they may be acting in a paracrine fashion, releasing transmitters/peptides locally, modulating other neurons, directly stimulating muscles through the basement membrane, or representing en passant synapses within the plexus. In some cases, we detect what we interpret to be interneurons, closely associated with the neural plexus, colabelled with DAPI (Fig 7H). We also find neurons with a more circular cell body morphology along the mid-proboscis that project anteriorly (Fig 7I). In this juvenile, bipolar neurons in the proboscis project posteriorly and appear to terminate at the proboscis base rather than extending further posteriorly (Fig 7B). In other transgenic juveniles, the population of bipolar neurons, right above the first gill slit in the anterior dorsal midline trunk, project anteriorly and terminate at the dorsal proboscis base in many of the animals that were imaged (Fig 7J).

Overall, based on the data from this synapsin transgene, the *S. kowalevskii* ectoderm contains a range of neural cell types including pseudounipolar, bipolar, multipolar, and multiciliated neurons in the proboscis and collar (Fig 7K–7N). However, by far, the most prevalent type of labeled neuron is the bipolar morphology observed throughout the animal. The abundant bipolar neurons are distinguishable from other types of cells in the ectoderm because of the distinctive axonal projections into the plexus.

**Location of transmitter release.** eGFP+ neurons from the transgene, DiI staining on fixed tissue, and NeuO labeling in live tissue, all reveal the presence of varicosities along the axons for most of the labeled neurons, most likely the site of transmitter/peptide release, possibly resulting in volume transmission or as en passant synapses. To further test this and to determine the localization of synapses/transmitter release throughout the nervous system, we designed a construct based on the previous synapsin:eGFP from Fig 7 but with the addition of

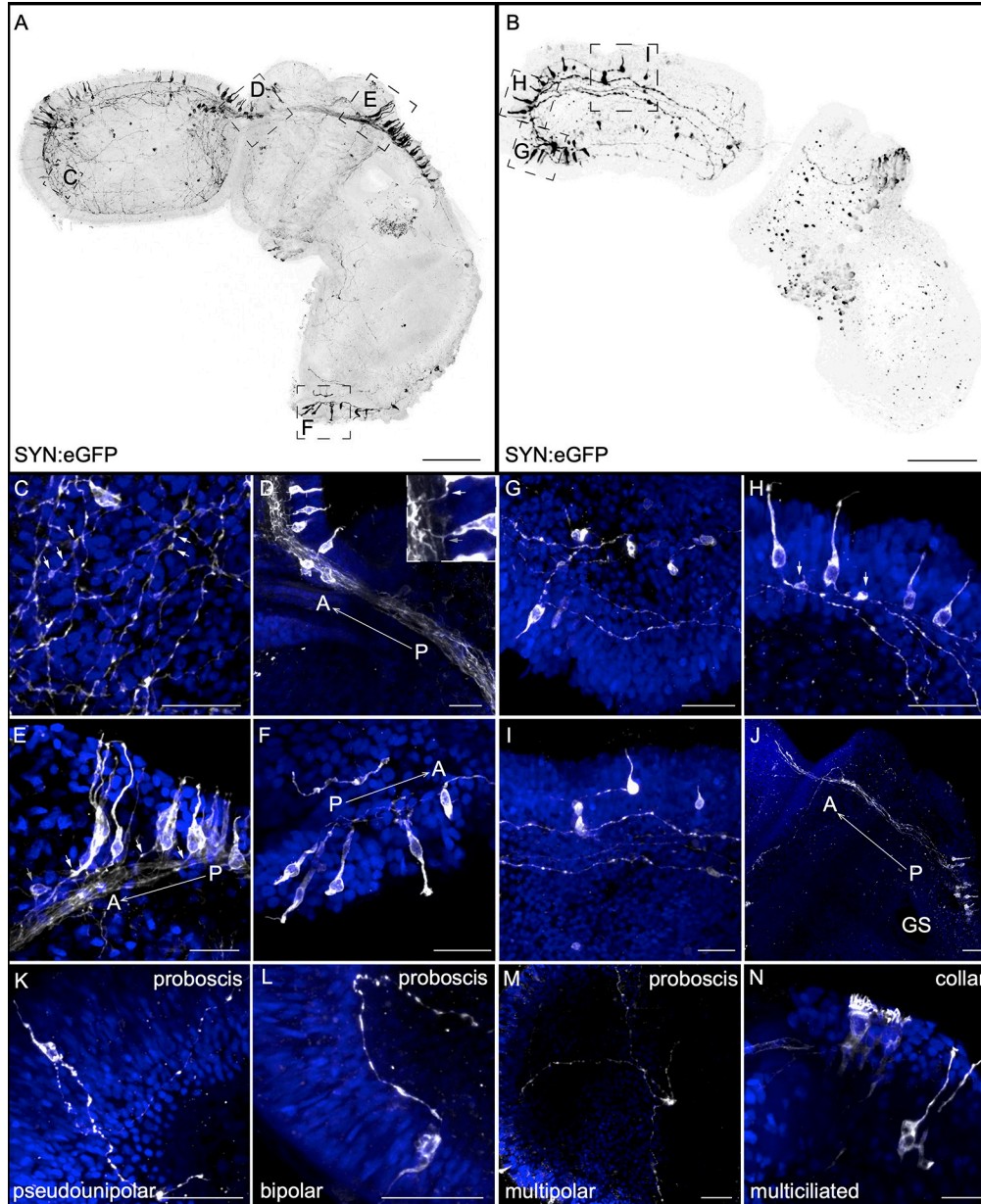

**Fig 7. Neural cell morphology and neurite projection using a synapsin:eGFP transgene.** (A, B) Maximum intensity projection of 2 representative synapsin:eGFP animals using 8 kb regulatory sequence upstream of the *synapsin I* gene to drive expression of cytoplasmic eGFP. (C-F) and (G-I) inset panels for each juvenile revealing cellular morphology and neurite projections. (C) The anterior proboscis plexus; white arrows point to varicosities along neurites. (D) The dorsal cord connecting the proboscis and collar; white arrow points to a single axon, and gray arrow points to a bifurcated axon from neurons projecting into the cord. (E) Dorsal posterior collar, neurons projecting anteriorly along the dorsal collar cord, with white arrows indicating projecting axons. (F) Postanal tail with detailed cellular morphology of neurons projecting anteriorly. (G) Anterior proboscis showing sensory cells projecting into the ectodermal plexus. (H) Pair of sensory cells in the apical tuft region that appear to project locally to interneuron in the plexus, indicated by white arrows. (I) Mid-proboscis with anteriorly projecting sensory neurons. (J) Projection trajectories from neurons in the dorsal posterior collar, right above the first gill slit, projecting anteriorly toward the dorsal proboscis base. (K-N) Representative neural cell type polarities across different animals. Transgene expression is in black in (A, B) and elsewhere in white with DAPI counterstained in blue. Scale bars in A, B represent 100 μm and 20 μm in Ai-G.

a mouse synaptophysin-mRuby fusion protein. Synaptophysin is a presynaptic vesicle protein [120,121], so we expect the fusion protein to be transported to either synapses or regions of vesicle release, as has been demonstrated in mouse using a similar construct [122]. The transgene generates mosaic eGFP expression similar to data presented in Fig 7 (Fig 8A and 8E). Within these eGFP-labeled neurons, we detect puncta labeled with mRuby, suggesting that the mouse Synaptophysin is trafficked successfully in hemichordates and labels regions of transmitter/peptide release. In the anterior proboscis (Fig 8B), there are many synaptophysin puncta along axons in addition to axon terminus containing a concentration of the synaptophysin fusion protein (Fig 8B–8D). We observe broad distribution of these puncta in both the anterior (Fig 8B) and posterior proboscis (Fig 8C). Synaptophysin also shows localization around the nucleus and along the cilium in bipolar neurons (Fig 8D). In one striking example, we were able to track a single axon from a cell body in the far posterior to the mid-proboscis (Fig 8E). Regular localization of Synaptophysin in swellings along the length of the axon supports our previous suggestion that the neuron is secreting transmitters/peptides locally or making direct neuron-to-neuron connections in the trunk plexus (Fig 8F–8H).

**Tyrosine hydroxylase transgene.** To investigate the cellular morphology and projections of catecholaminargic neurons, we designed a transgene using 5 kb of sequence directly upstream of the *TH* gene. Previously, we established that the cells expressing TH in the proboscis and collar also expressed the DAT, supporting the hypothesis that catecholaminergic TH-expressing cells are dopaminergic neurons (Fig 2). The TH:eGFP transgene expression is again expressed mosaically and mainly restricted to the proboscis, with a few isolated cells in the posterior collar (Fig 9A). Many of the labeled neurons in the proboscis have a flask-shaped morphology, as reported for synapsin:eGFP, and contain a single axonal projection directly into the plexus (Fig 9B, 9C, 9F and 9F'). In about half of the animals imaged, we identified a unique type of cell at the posterior collar and in many cases were able to trace their projections (Fig 9D and 9E). They have a unique asymmetric cellular morphology, a single dendrite with a terminal sensory cilium, and a protruding vacuole-rich mass. These cells have elaborate axonal trajectories across the collar and often project anteriorly into the proboscis (Fig 9G–9I'). Individual neurons were traced and 3D reconstructed using the 3D Visualization-Assisted Analysis (Vaa3D) software suite. TH transgene-labeled neurons in the anterior proboscis project posteriorly toward the proboscis base (Fig 9F and 9F'), whereas eGFP+ neurons at the proboscis base project anteriorly toward the proboscis tip (Fig 9J–9K'). Outside of the proboscis in the few eGFP+ neurons in the collar, neurons project anteriorly toward the proboscis base (Fig 9G–9I').

In summary, the transgenic data reveal the detailed cellular morphology of neurons across different regions of the body plan, and for the most part, neurons have a similar morphology, bipolar sensory neurons that project into the neural plexus and often across long distances. Fewer neurons with more diverse morphologies were described in classical studies.

## Discussion

### Distribution of neurons and neural cell types in *S. kowalevskii*

The expression of a wide range of molecular markers of neurons and neural subtypes in *S. kowalevskii* confirms the broad distribution of neurons in the ectoderm outlined in several classical descriptions [29,31,33,34]. However, unlike the simple neural plexus proposed by Bullock [31], our data suggest that the nervous system of enteropneusts is far from simple. The few molecular studies in hemichordate neural structure and organization have focused on the distribution of neurons in the cords, in particular the dorsal cord due to its proposed affinities with the chordate dorsal cord [54,123]. Our study provides additional insights by investigating

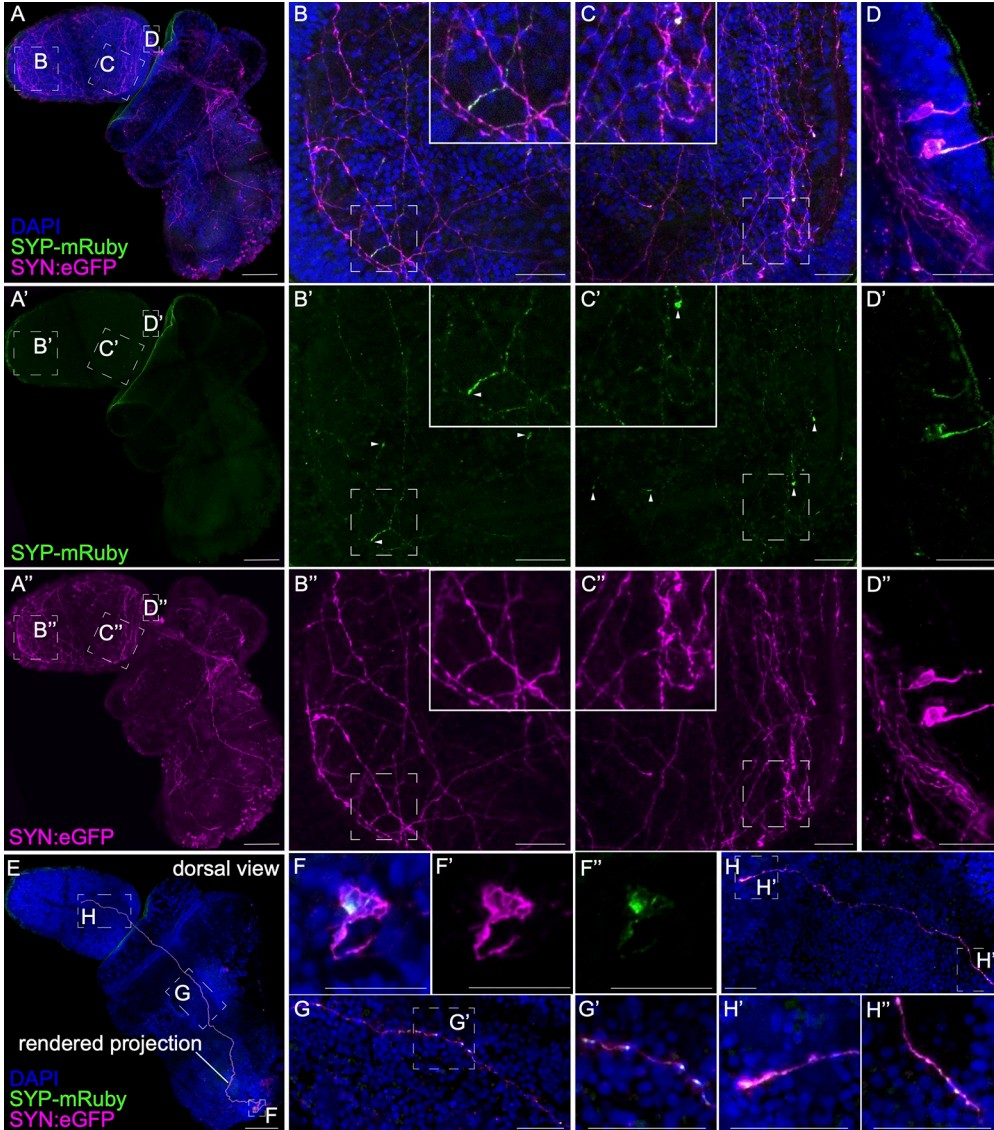

**Fig 8. Localization of synaptophysin along axons at varicosities.** (A) Mosaic expression of the synapsin:mGFP—T2A—(mouse)Synaptophysin-mRuby transgene in a representative animal. The transgene uses 8 kb regulatory sequence upstream of the *synapsin I* gene to drive expression of mGFP and the mouse synaptophysin-mRuby fusion protein cleaved off by the self-cleaving enzyme sequence, T2A. (B-D) Inset panels from (A) at different regions along the proboscis: (B) anterior proboscis, (C) posterior proboscis base, and (D) dorsal posterior ectoderm. (A'-D') Individual channel showing protein localization for synaptophysin-mRuby in green. White arrows in B' point to synaptophysin localization at axon terminals, and white arrows in C' point to protein puncta along the neurite. (A"-D") Individual channel showing protein expression for cytoplasmic eGFP in magenta. (E) Dorsal view of a second animal showing a single neuron expressing the transgene. The cell body is in the tail and projects into the proboscis. (F-H) Inset panels along the entire length of the neuron from (E): (F) neural cell body, (G) mid-axon at the dorsal posterior collar, (H) axon terminal at the mid-proboscis. (F-F") The panels show the (F) merged, (F') green, and (F") magenta channel at the cell body. (G', H', H") Inset panels from regions along the neurite showing SYP-mRuby puncta along the axon in G' and H" and at the axon terminal in H'. Scale bars are 100 μm in A-A", E and 20 μm in B-B", C-C", D-D", F-F", G-G', H-H".

the entire nervous system including both the cords and the extensive neural plexus. The data clearly identify a complex arrangement of spatially segregated neural subtypes that is most prominent in the general ectoderm rather than in either cord. The proboscis epithelium is the

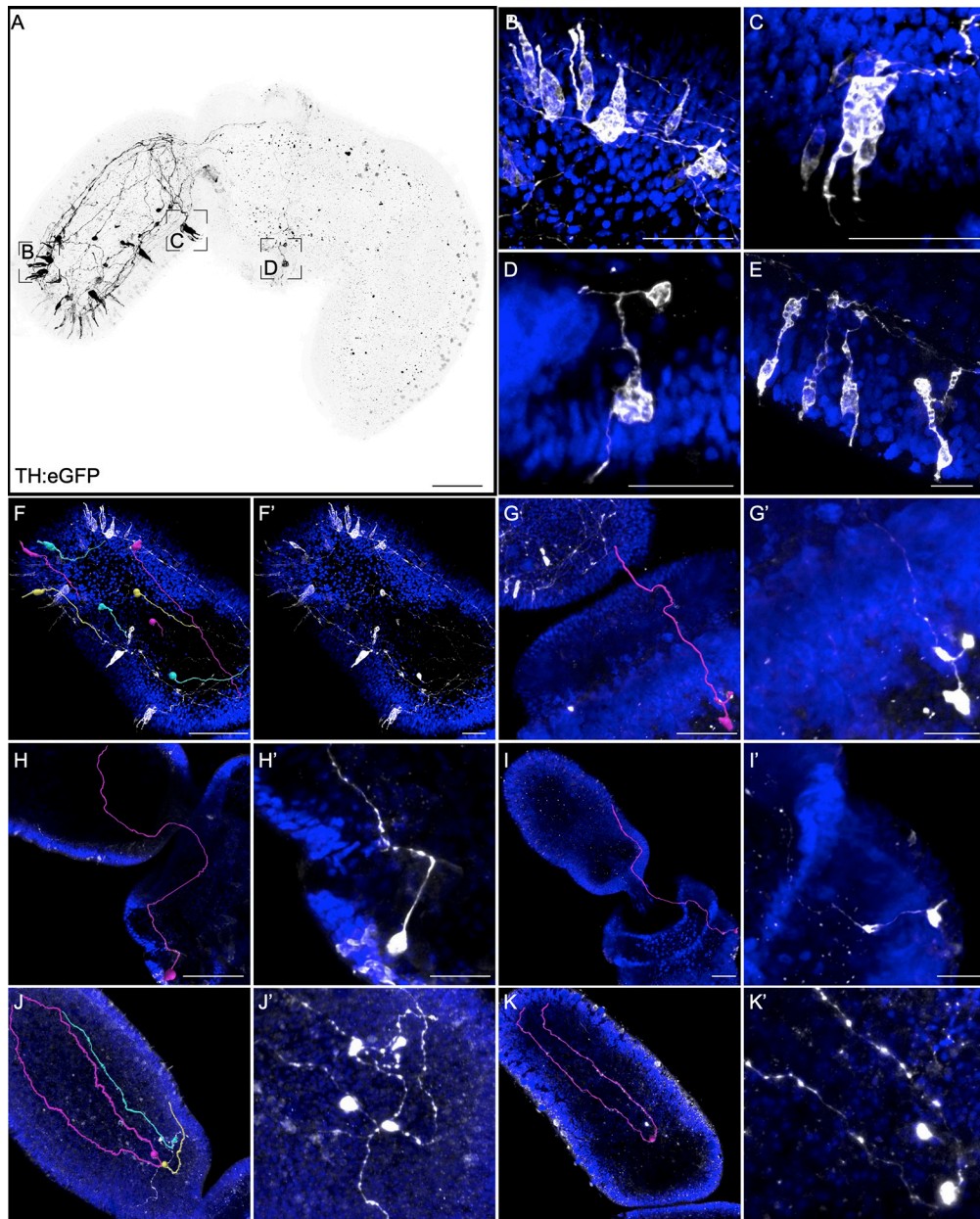

**Fig 9. Neural cell morphology and neurite projection using a TH:eGFP transgene.** (A) Maximum intensity projection of a TH:eGFP representative animal using 5 kb regulatory sequence upstream of the *TH* gene to drive expression of cytoplasmic eGFP. (B-D) Inset panels from (A) at the (B) anterior proboscis, (C) ventral posterior proboscis, and (D) ventral collar. (E) Proboscis ectoderm from a different animal showing detailed cellular morphology of TH+ neurons. (F-K') Manually traced axons from sensory neurons that project posteriorly in the proboscis (F, F'), collar neurons that project anteriorly toward the proboscis (G-I'), and proboscis base neurons that project anteriorly toward the proboscis tip (J-K'). Transgene expression is in black in (A) and elsewhere in white with DAPI counterstained in blue. Scale bars in A represent 100 μm, 50 μm in F-K, and 20 μm in B-E and F'-K'.

most richly innervated region of the animal, particularly at the base and close to the proboscis stem on the dorsal side, as has been described in other species of enteropneust [54]. The collar ectoderm is densely populated with neurons but the trunk far less, so neurons are also concentrated along the midlines in both the dorsal and ventral cords (Fig 1P and 1S). Data from *elav*,

widely used as a pan-neural marker, do not seem to label all the neural complement, as some neurons in the epithelium are not *elav*+, shown by the extensive expression of 5HT in the collar epithelium (Fig 5I).

The strongly regionalized distribution of neural subtype marker expression in the epithelium for neurotransmitters and peptidergic neurons suggests that there is marked differentiation of the nervous system in both the AP and DV axes (Figs 2 and 3). At the late juvenile stages, we observed strong regionalization of specific neural markers, densely packed and largely expressed in distinct rings, predominantly in the anterior plexus. However, we also saw evidence of clear molecular differentiation of neural subtypes between the dorsal and ventral cords, but in relatively few markers when compared with expression in the plexus. These data provide a broader view of neural cell type specification throughout the body rather than uniquely on cord differentiation [53,54,123].

## Cell type regionalization in the general ectoderm

We observed the most complex regionalized patterns in the proboscis and collar epithelium and far fewer in the trunk. However, far from the simple nervous system proposed by Bullock [31], our molecular analysis suggests a highly regionalized nervous system with a particularly complex organization at the base of the proboscis: There were 3 general expression domains: apically restricted, broadly expressed, and localized to the base, with some markers represented in multiple domains. Most of these neural populations show circumferential domains with little evidence of dorsoventral differentiation reflective of the structure of the plexus. The base of the proboscis is both the region of highest neural density and the most diverse in terms of neural cell types. Dopaminergic (DA), GABAergic, histaminergic, and peptidergic neurons show circumferentially localized domains of expression in this region (Figs 2 and 3), and as we also demonstrate, it is also one of the most distinctive regions of neural plexus organization (Fig 4). The expression of GABA in the proboscis of *P. flava* exhibited a similar distribution to our findings, although their focus was largely restricted to the base of the proboscis and stem [54]. In the collar, the diversity and density of neurons is less than in the proboscis, and in the trunk of late juveniles, the only 2 neural subtypes represented in the general ectoderm of late juveniles, not associated with the cords, are serotonin and DA.

The rings of neural subtypes in both the collar and proboscis ectoderm are very similar to the expression domains of the regulatory genes with conserved bilaterian roles in CNS patterning described in previous studies on *S. kowalevski* and other enteropneusts species [11,54,59]. What is particularly striking is the clustering of neural subtypes in the regions of the ectoderm that are the sites of localized epithelial signaling centers during early development. The apical tip of the developing proboscis is the site of active FGF and Hedgehog (Hh) signaling and a source of Wnt antagonists [46,49]. This territory has been compared to the vertebrate anterior neural ridge. We observe a wide range of neural subtypes clustered in this region. At early developmental stages, the boundary between the proboscis and collar is the site of a narrow circumferential stripe of transitory Hh expression that strongly resembles the regulatory gene expression profiles of the vertebrate zona limitans intrathalamica (ZLI). We notably observed a tight localization of GABA and DA neurons clustered in this region on either side of this organizer (Fig 2B' and 2G'). Finally, the boundary of the collar and trunk is the site of the expression of Wnt1 and FGF8, which are the characteristic ligands of the isthmus organizer at the midbrain hindbrain organizer [49]. This is a key organizer for the formation of the midbrain dopaminergic neurons in vertebrates, and both TH in situs and transgenics show TH neurons in this general region (Figs 2A, 2B', and 9A). Further functional tests will be required to determine whether these conserved regulatory networks are involved in the regulation of

specific neural subtypes in these territories. This raises the exciting possibility that the conservation of gene regulatory networks between these disparate body plans is related to their role in the positioning of conserved cell types along the AP axis.

## Origins of hypothalamus and pituitary

The clustering of neurons around the base of the proboscis that express orthologues of neuropeptides/neurohormones that are involved in the function of the hypothalamic/pituitary axis in vertebrates is of particular interest. Evolutionary insights into the origins of the neurosecretory centers of the vertebrate brain have come from Amphioxus and tunicates, but little is known outside of chordates [124,125]. Studies from the annelid *Platynereis dumerilii* have demonstrated a potential hypothalamic precursor, suggesting a deep ancestry of neurosecretory centers in bilaterians [7,89]. Echinoderms and hemichordates have largely been excluded from a broader synthesis, except for the early pioneering studies by Bateson [19] who compared the proboscis pore at the dorsal base of the proboscis to Hatschek's pit in Amphioxus linking it to pituitary and hypothalamic origins, and Komai [126] who compared the stomocord to the pituitary. The clustered expression of many of the orthologues of characteristic neuropeptides/neurohormones of the hypothalamus (CRH, CalC, Orexin, and TRH) around the base of the proboscis is very provocative as this region of the plexus overlies the heart/kidney complex of the worm. A more rigorous characterization of this region is now warranted to investigate whether the projections of the neurons expressing neurohormones project to a similar region and whether this territory represents a basic neurosecretory center, releasing neuropeptides into the underlying coelomic fluid or circulatory system. Strikingly, the homeobox gene *pitx* (pituitary homeobox) is expressed in a prominent spot at the base of the proboscis [48] and suggests that a thorough description of regulatory gene expression and function involved in vertebrate pituitary and hypothalamus morphogenesis could further help establish a comparative basis for neural anatomical comparisons [127].

## Cell type regionalization in the cords

Both the dorsal and ventral cords show characteristic expression domains of specific neural cell types. However, we detect the expression of only a subset of the markers that we see prominently expressed in the proboscis. Despite fewer neural subtypes, those that are expressed show that the cords are distinct, suggesting differentiation of function. Given our analysis was largely restricted to late juveniles, it is possible that the complement of neural subtypes expands as the animals grow larger. The dorsal cord is divided into the internalized collar cord and the superficial cord that runs the length of the trunk to the anus. We see little evidence of any neurotransmitter marker expression in the collar cord soma, except for peptidergic neurons, despite the prominent expression of *elav*. In the more posterior domains, where the cord is superficial and basiepidermal along the trunk, there are glutamatergic neurons on either side of the dorsal cord and few isolated GABA-labeled cells in its most anterior extent. Both cords show clear expression of peptidergic neurons based on the expression of the processing enzymes and several neuropeptides. The length of the ventral cord shows strong and broad expression of the histidine marker HDC (Fig 2E).

When the expression of a wide range of markers of neural differentiation are considered, in both the epidermis and in the cords, we see that most neurons at the late juvenile stage are in the proboscis ectoderm rather than in the cords. The region of the highest neural density and diversity is in the proboscis ectoderm, around the base. A previous study in *P. flava* [54] had identified the proboscis stem as a region with strong neural differentiation, we confirm this, but we would include the entire proboscis base in this region of rich neural diversity rather

than just the dorsal territory. Many of the gene expression domains in this region are organized in circular rings reminiscent of the expression of nested transcription factors with key roles in neural cell fate determination in vertebrate brain development.

## General organizational features of the plexus and cords

Previous studies have revealed the pervasive plexus present throughout the epidermis, and our data further refine the structural details of the plexus adding significant details [27,28,31,34,128]. The plexus is more prominent in the proboscis and with a striking change in organization at the proboscis base, where its organization transitions from parallel nerve bundles running along the AP axis into a mesh of axonal bundles containing both serotonergic, GABAergic, and peptidergic processes (Fig 4). A very similar plexus structure has recently been reported in different acoel species [109]. In these cases, it is reported that the reticulated neurite bundles wrap around cell clusters. Dilly [35] reported that the enteropneust epithelium is strongly bound to the basement membrane by cellular processes that penetrate this plexus, so it is possible that this territory has its distinctive morphology due to cells penetrating and attaching to the basement membrane. The plexus is much thicker at the proboscis base and is most prominent on the dorsal side and possibly represents a center of integration [54]. While this structure is most striking at the base of the proboscis, a looser mesh of axonal bundles is also present in the collar. Confocal z-stacks of the plexus, stained with neural antibodies or following DiI labeling, reveal a clear apico/basal structural organization throughout the proboscis and collar plexus, indicative of function partitioning. This scenario explicitly contradicts classical studies [31].

Data from the transgenic embryos give particularly valuable insights into the structure of the plexus, as the mosaic incorporation of the transgene labels only a subset of neurons and enables the tracking of individual neurites. We see no evidence of a plexus organization like that of the simple nets of cnidarians, in which local projections synapse directly to neighboring neurons. Instead, we largely observe long-range projections from sensory neurons in the epithelium. Neurons in the anterior proboscis mostly project posteriorly to the base of the proboscis. However, there is a wide range of axonal trajectories observed, and at the base of the proboscis, we mostly see lateral projections but also some projections anteriorly. The few cells labeled in the trunk project anteriorly into the proboscis. In some experimental embryos, we were able to trace the neurite from the tip of the tail all the way up to the base of the proboscis, suggesting very long-range communication in the animal (Fig 6E). We were unable to find a single instance of the axonal projections crossing the basement membrane and into the muscles from either immunohistochemistry or transgenic data. Although it is possible that cholinergic neurons cross the basement membrane to innervate muscle, the general neuronal transgene, synapsin:eGFP, should have labeled a wide range of neuronal cell types including cholinergic neurons, yet we find no evidence for neuron-to-muscle innervation. This finding is significant as there has been some disagreement in the classical papers as to whether there is direct innervation of muscles and evidence of axonal processes crossing the basement membrane, summarized most recently by Dilly [35]. Light microscopy reports were decidedly mixed in their assessments with some confirming axonal crossings, but mostly were cautious and reserved in drawing conclusions [31,34,128,129]. In Hyman's invertebrate treatise, she concluded that there was no compelling evidence for fibers crossing the basement membrane [130]. With the advent of electron microscopy, this issue was revisited: Two independent studies suggest direct innervation of muscles from fibers crossing the basement membrane [35,131], but neither study report evidence of neuromuscular junctions. This question would clearly benefit from further clarification using modern volume EM approaches [132].

Comprehensive examination of the sister group to hemichordate, the echinoderms, has shown a similar lack of direct muscular innervation and penetration of the basement membrane by neural fibers in any echinoderm group examined [112,133,134]. Even in chordates, Amphioxus, the most basally branching chordate lineages, muscle fibers extend to the neural tube and stimulation occurs at this basement membrane interface [135], suggesting that sophisticated animal behavior can be mediated without a direct neuromuscular junction.

Our data further provide insights into the potential predominant mode of neural transmission in *S. kowalevskii*. The transgene and DiI data are particularly informative: All axons observed at high magnification were characterized by regular varicosities along their length. One striking synapsin transgenic embryo showed regularly spaced varicosities along the axon of a single neuron from the tail all the way up into the base of the proboscis (Fig 8E–8H). Synaptophysin protein localized to the varicosities and suggest that transmitter/neuropeptide release occurs along the length of the neuron. Varicosities were also pervasive in the proboscis neurons, lending support to the idea that, like in echinoderms, communication may largely be paracrine across the plexus, with general transmitter/neuropeptide volume release. Volume transmission is recognized as a critical component in a wide range of neural systems from the simple to the more complex [112,136–139]. In many extant nervous systems, from vertebrates to *C. elegans* and marine larvae, paracrine, chemically wired brain centers are integrated with synaptic networks, and some authors have proposed that most extant nervous systems likely evolved from peptidergic, paracrine systems [140]. We propose that during the early ambulacrarian evolution, paracrine signaling was evolutionarily favored as the predominant mode of neural transmission. We cannot rule out the presence of true synapses between neurons in the plexus and the cords, but at least in the juveniles we examined the presence of neuromuscular junctions is unlikely, and we saw no evidence of the presence of axonal processes crossing the basement membrane into the muscle as was reported by early EM studies [35]. However, it is possible that these develop as the animals grow larger. Nieuwenhuys from his review of volume transmission across animal groups [112] makes the key statement that "the notion that volume transmission is primitive, generalized, sluggish and lacks precision, whereas wiring transmission is advanced, specialized, fast and accurate, is erroneous." Our data need to be corroborated by neurophysiological assays to determine the function of the nervous system in enteropneusts but already set up some clear hypotheses that can be further tested by some targeted volume EM to investigate the detailed structure in the plexus at a variety of regions of the body plan. Of course, this is a description of a single species representing one family of enteropneusts. Follow-up studies in additional species, representing a broader range of diversity, will be required to determine whether the details of *S. kowalevskii* neuroanatomy adequately represents the main organizational features of enteropneusts.

## Central or decentralized nervous system

Much of the comparative interest in enteropneusts has been in its potential to provide insights into the early origins of chordates. The structure of the nervous system is a critical character for inferring ancestral states of early deuterostome nervous system [10]. Our data provide some insights into this question but also raise many others. A simple statement to satisfy whether enteropneusts are characterized by a CNS clearly delineated from a PNS seems largely driven by our expectations based on studies from highly derived, cephalized model animals. Our data do not provide a clear answer to the question.

The transgenic animals are the most informative on this issue as they reveal that the predominant neural cell type is a bipolar sensory neuron that project across long distances, both in the proboscis and in the trunk. We found scant evidence of the multipolar neurons

described in classical studies at the basal side of the plexus. In a few animals, we identified what we interpreted as a potential interneuron close to the predominant bipolar sensory neurons in the tip of the proboscis and a few instances of multipolar neurons in the proboscis plexus (Fig 7). From our data so far, it is difficult to delineate a distinction between a clearly defined CNS and a PNS. A previous molecular study [54] concluded that the dorsal cord and proboscis stem represented a "bona fide CNS," and the rest of the proboscis and remaining epithelium was a PNS. Our data do not support such a strong division, as most neurons identified are sensory, with few candidates for interneurons. At least at this juvenile stage, a division between peripheral and central is not obvious, and the varicosities throughout the plexus are likely involved in processing sensory information. Our data do support an important role of the dorsal proboscis stem as a region of potential integration [54], as we observe many axonal processes from transgenic embryos projecting to this region, consistent with previous hypotheses, but the entire proboscis base might be a region of integration. We find little support for a special role of the dorsal cord as an integrative center, but perhaps the cords take on more significant roles in processing as the animals grow. However, the characterization of the rest of the proboscis as a PNS separated from a central nervous system is not well supported. Our transgene data support transmitter release throughout the plexus that may suggest information processing occurs throughout the plexus of the animal. The broad dispersal of many neurons, characterized by peptidergic and classical transmitters, particularly in the proboscis and collar, also suggests that signal propagation occurs through tiling in a manner that has been proposed to represent ancestral nervous system function [140], and the characterization of the animal as having a "skin brain" might be the best analogy [141]. Whether these varicosities represent true synapses or regions of volume release will require more detailed analysis with EM.

In conclusion, these data further build on existing research from classical studies in demonstrating that hemichordates do not share many organizational principles to the canonical centralized nervous systems of the main models that we study in neurobiology. This study in enteropneusts demonstrates that we have much to discover by sampling a far more diverse array of neural systems representing the rich biodiversity of marine environment. Our molecular characterization does not reveal cryptic similarities with chordates that were not apparent from previous descriptive work. Yet, despite these profound differences in neural architectures, the early ectodermal patterning that establishes the contrasting neural systems of vertebrates and hemichordates is highly conserved [11,49,50]. This work furthers the counterintuitive observation that regulatory conservation between distantly related groups has seemingly not restricted morphological diversification to a specific neural conformation over macroevolutionary time frames. We are far from understanding the link between gene regulatory conservation and nervous system evolution, and only by broadening our molecular scope into biodiversity are we likely to be able to recognize cryptic links between morphological and molecular evolution that will allow us to address these important but difficult questions of nervous system origins.

## Materials and methods

### Animal collection and embryo culture

Adult worms were collected in the months of May and September during the *S. kowalevskii* breeding seasons in Waquoit Bay, MA, and maintained in flow-through sea tables at the Marine Biological Laboratory in Woods Hole, MA. Spawning fertilization and embryo culture followed protocols developed by Colwin and Colwin [142,143] with updated methods [144].

## Cloning of orthologs

*S. kowalevskii* homologs of vertebrate genes were identified in an EST library screen [145]. See S1 Table for NIH accession numbers.

## Colorimetric in situ hybridization

Colorimetric whole mount in situ hybridization on juveniles was carried out using an established lab protocol [144]. Embryos were kept in 5 mL glass vials for all steps until the colorimetric reaction performed in 6-well tissue culture plates. Samples were fixed in 4% paraformaldehyde (PFA). Proteinase K treatment was carried out at 10 µg/mL in PBST (0.1% TritonX in 1x PBS) for 15 minutes at room temperature (RT). Acetic anhydride treatment at 250 µM for 5 minutes at RT followed by a 500-µM treatment for 5 minutes at RT. In situs for experimental embryos were stained using 1.6 µL:2.7 µL ratio of 5-Bromo-4-chloro-3-indolyl phosphate: nitro-blue tetrazolium chloride and stopped with $3 \times 5$-minute rinses at RT in 1XMAB. Some samples were further cleared by rinsing $2 \times 5$ minutes in MeOH and cleared with a 2:1 ratio of benzyl benzoate:benzyl alcohol (BBBA) before imaging on a Zeiss Axioimager.

Whole mount in situ hybridizations on adults up to 4 gill slits utilized the same protocol as embryos with minor modifications: Samples were permeablized for 15 to 20 minutes at RT in Proteinase K diluted 1:7,500 in 1X PBST to increase probe penetration and staining in deeper tissues. The proboscis coelom and gut were punctured with a tungsten needle or scalpel blade to reduce probe trapping and increase penetration. For adult whole mount in situ hybridization, samples were dissected using a scalpel or scissors to reduce probe trapping and increase staining of deeper tissues. The whole mount in situ hybridization protocol was modified for adults by utilizing either large glass vials or 15 mL tubes and performing at least 2 more washes than embryos for all steps.

In situ hybridization on tissue sections was based on the protocol for embryos with technical modifications for slides. Adults were fixed in 4% PFA and then dissected into pieces no more than 3 cm in length and cryoprotected in an increasing concentration gradient of sucrose in fixation buffer (3.7% formaldehyde, 0.1 M MOPS (pH 7.5), 0.5 M NaCl, 2 mM EGTA (pH 8.0), 1 mM $MgCl_2$, 1X PBS) up to 20% sucrose at RT and allowed to equilibrate overnight at 4°C. Samples were then placed in 20% sucrose in fixation buffer diluted 2:1 in OCT media (Fisher) and allowed to equilibrate at RT with gentle agitation. Fixation buffer was utilized instead of 1x PBS to reduce tissue swelling and sectioning artifacts. Samples were embedded in plastic molds, chilled on dry ice, and stored indefinitely at −80°C. Frozen blocks were sectioned at 16 to 25 mm using a Leica cryostat with a cutting temperature between −25 and −30°C. Sections were collected on Superfrost Plus slides (Fisher), allowed to dry for at least 20 minutes, and either processed for in situ hybridizations or immunofluorescence immediately or stored at −80°C. Sections were permeablized with 0.2% Triton X-100 in 1X PBST for 20 minutes rather than with Proteinase K, and all washes were performed in upright slide mailers. For hybridization, sections were covered using plastic Hybrislip cover slips (Grace Technologies) and placed facing up in horizontal slide mailers with a Kimwipe or filter paper soaked in hybridization solution to prevent the slide from drying. Slide mailers were sealed using tape, placed in chambers constructed from empty pipette tip boxes, and hybridized overnight in an oven at 60°C. For blocking and antibody steps, sections were outlined using a hydrophobic marker, and incubations were performed on slides placed horizontally in humidified chambers at RT or at 4°C overnight. Different probes were hybridized in separate chambers and imaged on a Zeiss Axioimager.

## HCR in situ hybridization

DNA probe sets were generated and ordered from Molecular Instruments using the full RNA sequence from GeneBank accession numbers TH (XM_006813504.1), DAT (NM_001168055), VGLUT (XM_002739644), GAD (XM_002740628), achatin (XM_002732101.2), CCK (XM_002738068.2), and orexin (XM_002734948.2). For each gene, 11 to 33 probe sets were generated, and DNA oligo pools were resuspended to a final concentration of 1 μmol/μl in 50 mM Tris buffer (pH 7.5). HCR amplifiers, B1-Alexa Fluor-546, B2-Alexa Fluor-488, and B3-Alexa Fluor-647 were ordered from Molecular Instruments. We followed the HCR version 3.0 protocol [146], with some minor modifications using a protocol from Nipam Patel's lab [147].

## Immunohistochemistry

Juveniles were reared and fixed as previously described [144]. Briefly, embryos were fixed for 30 minutes at RT in fixation buffer (3.7% formaldehyde, 0.1 M MOPS (pH 7.5), 0.5 M NaCl, 2 mM EGTA (pH 8.0), 1 mM MgCl2, 1X PBS) and subsequently stored in ethanol at −20°C. For anti-GABA, we used 3.7% formaldehyde and 0.3% glutaraldehyde. For antibody staining, embryos were rehydrated into 1x PBS + 0.1% Triton X-100 (PBST), rinsed 3 × 10 minutes in PBST, and placed into a blocking solution of 1x PBS +0.1% Tween 20 (PBT) +5% goat serum for 2 hours at RT. Embryos were incubated in PBT with either anti-GFP (Life Technologies, #A-6455), anti-FMRFamide (Immunostar, #20091), anti-5HT (Sigma, S5545), or anti-GABA (Sigma, A2052) at a 1:500 dilution overnight at 4°C. After primary antibody incubation, embryos were washed 4 × 30 minutes in PBST and then incubated for 4 hours at RT with secondary antibody (Alexa-Fluor 488 goat anti-rabbit IgG, Thermo Fisher #A-11008) diluted 1:500 in blocking solution. Samples were then washed 4 × 30 minutes in PBST and cleared into 80% glycerol. Some samples were further cleared by rinsing 2 × 5 minutes in MeOH and cleared with a 2:1 ratio of BBBA. Images were captured on a ZEIS LSM 700 confocal microscope with 20× and 40× objectives using the Zen software package (Carl Zeiss).

Immunofluorescence on adult tissue sections was performed as described previously with minor modifications. Sections were outlined with a hydrophobic marker, and wash steps were performed in an upright slide mailer. Blocking and antibody incubation steps were performed on slides placed horizontally in humidified chambers at RT. Primary antibody incubations were as follows: 1E11, 1:3 (gift of Robert Burke); rabbit anti-serotonin, 1:250 (Sigma S5545); mouse anti-FMRFamide, 1:600 (Immunostar #20091). Secondary antibody (Molecular Probes) dilutions were as follows: Alexa Fluor 488 goat anti-mouse or rabbit, 1:500; Alexa Fluor 546 goat anti-mouse or rabbit, 1:1,000; Alexa Fluor goat anti-mouse or rabbit 647, 1:250. Imaging was performed using a Zeiss LSM700 confocal microscope with Zeiss Zen software, an Olympus FV1000 confocal microscope with Olympus software, or a Zeiss Axioimager.Z1 compound microscope or Discovery.V12 stereomicroscope with an Axiocam MRm camera and Zeiss Axiovision software.

## Transgenes

Between 5 and 8 kb of sequence directly upstream of the start codon for genes *TH* (5kb, XM_006813504.1) and *synapsin* (8kb, XP_006820290.1) were cloned into a reporter plasmid based on the construct design in Lampreys [148] containing I-SceI meganuclease restriction sites, upstream of an eGFP coding sequence, and a SV40 late polyadenylation signal sequence using Gibson assembly [148,149] and previously published [119]. Full regulatory sequence for each transgene is included in S1 Table. Injection mixtures contained 10 μl restriction digest including 5 units of I-SceI enzyme (NEB), 1 μl CutSmart buffer, and 130 ng of reporter

plasmid, final concentration 13 ng/μl. The mixture was incubated at 37˚C for 40 minutes and injected into embryos between 4 and 9 minutes postfertilization as previously described [119]. To visualize regions of transmitter release, the mouse synaptophysin-mRuby fusion protein was cloned into the generated synapsin:eGFP transgene from AAV-FLExloxP-mGFP-2A-synaptophysin-mRuby (Addgene Plasmid# 71760) plasmid vector courtesy of Liqun Luo [122] to generate the synapsin:mGFP-2A-(mouse)synaptophysin-mRuby transgene. Vector inset sequence and primers used for the generation of these plasmids are listed in S1 Table. Reconstruction of neural projections for a subset of neurons were generated using the 3D Visualization-Assisted Analysis (Vaa3D) software suite [150].

## Supporting information

**S1 Table.** (A) NIH EST accession numbers for *S. kowalevskii* homologs of vertebrate genes identified in an EST library screen [145]. (B) Gene sequence and accession number used to generate the HCR hairpins. (C) Primer sequence used to generate fragments for transgene listed in D. (D) Transgene name and sequence used in this study as well as the gene accession number.
(XLSX)

**S1 Movie. A z-stack of an adult at the anterior collar clearly showing the layered nature of the plexus against the *synaptotagmin 1* (1E11) antibody.**
(AVI)

**S2 Movie. A z-stack along the collar and trunk of a 3GS juvenile.** The 5HT antibody clearly shows the serotonergic nervous system composed of many sensory neurons at the collar and trunk with a basipethelial neural plexus.
(AVI)

## Acknowledgments

We would like to thank the Staff of the Marine Biological Laboratory for hosting us and their assistance; The Waquoit National Estuarine Research Reserve for allowing access to our collection site; Jim Mcilvain from Zeiss for assistance in microscopy; Robb Krumlauf and Hugo Parker for providing the vector for our transgenics; Thurston Lacalli for his guidance and discussions on the evolutionary implications of this work; and Laurent Formery for his editorial input.

## Author Contributions

**Conceptualization:** José M. Andrade López, Ariel M. Pani, Christopher J. Lowe.

**Data curation:** José M. Andrade López, Ariel M. Pani, Mike Wu, John Gerhart.

**Formal analysis:** José M. Andrade López, Ariel M. Pani.

**Funding acquisition:** Christopher J. Lowe.

**Investigation:** José M. Andrade López, Ariel M. Pani.

**Methodology:** José M. Andrade López.

**Project administration:** José M. Andrade López, Christopher J. Lowe.

**Resources:** Mike Wu, John Gerhart, Christopher J. Lowe.

**Supervision:** Christopher J. Lowe.

**Visualization:** José M. Andrade López.

**Writing – original draft:** José M. Andrade López.

**Writing – review & editing:** José M. Andrade López, Ariel M. Pani, John Gerhart, Christopher J. Lowe.

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
