## [Editor Report · Decision Letter 0]

9 Apr 2023

Dear Dr Andrade Lopez, 

Thank you for submitting your manuscript entitled "Molecular characterization of nervous system organization in the hemichordate Saccoglossus kowalevskii" for consideration as a Research Article by PLOS Biology.

Your manuscript has now been evaluated by the PLOS Biology editorial staff, as well as by an academic editor with relevant expertise, and I'm writing to let you know that we would like to send your submission out for external peer review.

IMPORTANT: After discussion with the academic editor, we think that it might be best if this paper were reviewed as a Resource paper. No re-formatting is required, but please select "Methods and Resources" as the article type when you upload your additional metadata (see next paragraph).

Once your full submission is complete, your paper will undergo a series of checks in preparation for peer review. After your manuscript has passed the checks it will be sent out for review. To provide the metadata for your submission, please Login to Editorial Manager (https://www.editorialmanager.com/pbiology) within two working days, i.e. by Apr 11 2023 11:59PM.

Kind regards,

Roli Roberts

Roland Roberts, PhD

Senior Editor

PLOS Biology

rroberts@plos.org

---

## [Decision Letter · Decision Letter 1]

18 May 2023

Dear Dr Andrade Lopez,

Thank you for your patience while your manuscript "Molecular characterization of nervous system organization in the hemichordate Saccoglossus kowalevskii" went through peer-review at PLOS Biology. Your manuscript has now been evaluated by the PLOS Biology editors, an Academic Editor with relevant expertise, and by three independent reviewers.

You’ll see that reviewer #1 says that your paper describes a valuable resource, but that it suffers from “poor labelling of the figures and unclear or ambiguous anatomical descriptions or figure annotations” – s/he supplies a very long list of the specific presentational problems. Reviewer #2 is extremely positive and has no requests. Reviewer #3 is also very positive, but thinks there are presentational problems – s/he suggests shortening and/or re-structuring the paper, and breaking some Figures up. This reviewer has attached an annotated version of the manuscript PDF, covered in helpful notes and suggestions.

In light of the reviews, which you will find at the end of this email, we are pleased to offer you the opportunity to address the comments from the reviewers in a revision that we anticipate should not take you very long. We will then assess your revised manuscript and your response to the reviewers' comments with our Academic Editor aiming to avoid further rounds of peer-review, although might need to consult with the reviewers, depending on the nature of the revisions.

**IMPORTANT - SUBMITTING YOUR REVISION**

*Resubmission Checklist*

*Published Peer Review*

*PLOS Data Policy*

*Blot and Gel Data Policy*

Sincerely,

Roli Roberts

Roland Roberts, PhD

Senior Editor

PLOS Biology

rroberts@plos.org

REVIEWERS' COMMENTS:

Reviewer #1: 

This paper provides a detailed description of the anatomy of the nervous system in juveniles and adults of the hemichordate Saccoglossus kowalevskii nervous system using in situ hybridization, immunostaining and transgenic reporters for multiple neuronal markers (including neurotransmitters and neuropeptides). Although various aspects of hemichordate NS organization have been previously described, the present study provides the most comprehensive overview of the hemichordate NS to date and reveals new insights into its organization, for example by highlighting several unexpectedly long range connections between cell populations. The paper will therefore present a valuable resource for anyone interested in the evolution of nervous systems. 

In its current stage, however, the paper suffers from poor labelling of the figures and unclear or ambiguous anatomical descriptions or figure annotations, which makes it difficult for the uninitiated reader to follow the descriptions. This necessitates a thorough revision of the text and figures as suggested in my detailed comments below.

Detailed comments:

- 57: delete "now"

- 79/80: one ventral cord and one dorsal cord

- Fig.1: 

* label sections in A with panel numbers; 

* explain all abbreviations (e.g. PDC, PB, PS); 

* the placement of labels often is ambiguous - use arrows or lines to disambiguate eg it is not clear which structure is ACR); 

* always indicate orientation of sections (eg. where I dorsal in B, C?); I

* 'm confused about the staining domains in D: there are two domains in the ectoderm on top, one at the bottom, what are they?; 

* K-M; label structures in sagittal sections: 

* N,O: label proboscis, collar and collar cord

- 129: "expression of GRN" is awkward wording

- 130-132: almost nothing is known about regulatory links in GRN of Saccoglossus

- 196: where are elav cells in gut? Label in Fig. 1!

- 215: It needs to be stated here that all the following data are based on in situ hybridization and table Suppl. 1 needs to be cited. The wording in this section often suggests misleadingly that you describe the distribution of neuropeptides, where you really describe the expression of genes encoding their unprocessed precursors. Explain whether each of these genes encodes only a single neuropeptide. Please also explain why some of the staining in Fig 2-3 appears as NBT/BCIP signal, while other staining appears fluorescent (HCR?) 

- Fig.2:

* You need to clarify that these are in situs; 

* explain briefly which panels are based on HCR or on NBT/BCIP;

* explain whether blue counterstain is DAPI;

* all abbreviations must be explained 

* C'': many TH cells do not coexpress DAT

* C''': TH and DAT staining barely visible

* What are open arrows in F/F'?

* What are we supposed to see in H?

- 244:: "TH cells at the base of the proboscis". It is not clear which cell group in Fig. 2 you mean; no reference to Fig. panel

- 300: "dense anterior expression": this is not visible in Fig. 2H

- 336: delete "P"

- Fig. 3:

* Again, emphasize that we see in situs here (and which kind of staining)

* The legend is written in a confusing way; please follow the following template: "(X, X'): Expression of … in early (X) and late (X') juvenile"

* Inset in G' is smaller than main panel but shows the same view, so please remove

* I': how can you see the ventral cord in a dorsal view?

* What do the arrowheads in J, K, Q mean?

* N': if inset shows "RNA-based probe" what is shown in main panel?

* N-O: location/orientation of insets unclear

* L: Corticotropin (not -trophin)

* R-R'': not clear , which, if any, cells are double labeled; show individual channels also

- 365: why is vasotocin not shown next to orexin?; they are discussed together in text

- 376: gill slits and ciliary band should be labelled in Fig. 3 J'

- 409: posterior projections not clearly visible in Fig. 3 N

- 441: with a monoclonal

- Fig. 4:

* Again it is not clear what we see here; if all of these panels are based on fluorescent immunostaining, why is the signal black in A and white in B etc?

* Label anterior collar ring and the multiple structures visible in collar in A

* For clarity, I would suggest to discuss this overall organization of the nerve tracts earlier in the manuscript, following the description of elav

* Is the description for D and E switched? Where is the dorsal cord? What are the circumferential fibers in D?

* F: if synaptotagmin antibody labels neurites , why can we not see individual neurites here?

* J,K,L,M: orientation unclear

* J,K: what do the arrows mean?

- 476: I cannot see the "regularly spaced connectors"

- 478-483: I cannot follow the description here. The circumferential neurites in Fig. 4 D are not described. I don't understand what is meant by "series of layered, interconnected tracks"

- I could not run the supplementary movies, so cannot comment on their quality

- "The results from 1E11": awkward wording

- 495: please discuss why there may be "little resolution of individual neurites"

- Fig. 5:

* Label gill slits in A

* B: from which region is this?

* D: why is anterior ring not included here?

* I,J Orientation unclear; is the proboscis removed in I?

* K this shows transverse and not sagittal sections

- 557/558: distribution of

- 583: indicate location of endodermal plexus in the figure

- 600: Fig 6Aiii (not 56iii)

- Fig.6

* In previous figures, panels showing details of A have either been labelled B,C… or A', A''; here now they are labelled Ai, Aii… - please be consistent

* A': please label gill slits

* B-E, G: orientation unclear

- 617: anterior ring not visible/labelled in Fig. 6C

- 663/664: please add parentheses and Fig. number to "white arrows in inset" and "grey arrows in inset"

- 672: repeats previous statement in 666

- Fig. 7: 

* Explain why signal black in some panels, white in others

* Is blue staining DAPI?

* There are no re arrows only white and grey and they need to be explained

* Box for I in panel A is missing

* D: Spelling "Pseudounipolar"

- Fig.8:

* Labeling of panels (A,B,C…) is missing

* Arrowheads not explained

- Fig. 9: Is blue staining DAPI?

- 773: replace "with " with "were"

- 795: reword sentence "epithelium of … neurotransmitters…"

- 802: delete "plan"

- 825: delete "and"

- 840-856: The discussion here makes no sense to me. In vertebrates the posterior pituitary forms as an extension from the hypothalamus but the anterior pituitary from the adenohypophyseal placode, while the hypothalamic GnRH neurons derive from the olfactory placode. Which regions of the hemichordate NS are here proposed to be homologous to what? Please note also that TH cannot be considered a specific hypothalamic marker (and I'm not sure about calcitonin).

- 910: cholinergic neurons were not described here. Is it possible that motor neurons, which innervate the muscles and possibly penetrate the basement membrane are cholinergic?

- 1020: what were the NBP/BCIP concentrations?

Reviewer #2:

In this study, the authors present the first molecular characterization of the Saccoglossus kowalevskii nervous system, using a combination of RNA in situ hybridizations, transient transfections with fluorescent reporters, and immunostainings. The images are wonderfully detailed and the text is well-written and thorough. It is clear that this paper is already highly polished. This represents an important characterization of a nervous system in a key branch of deuterostome phylogeny. It is therefore of great interest to a broad readership. I truly do not have any substantial criticisms or comments to be addressed. I fully recommend its publication in PLoS Biology.

Reviewer #3:

[IMPORTANT: This reviewer has attached an annotated version of your PDF]

The manuscript by Andrade Lopéz et al. provides a detail atlas of the nervous system of a hemichordate, a deuterostome animal of key relevance in understanding the evolution of the chordate nervous systems. This highly detailed anatomical of high quality study provides new insights into the organisation of the nervous system of these organisms. The authors carefully describe their findings and provide a balanced discussion of their relevance. The paper will be of interest to those interested in evo-devo, emerging marine animal models and the evolution of nervous systems.

The study uses a pan-neural marker and several neuron-type specific markers labeling various neurotransmitter or neuroendocrine pathways and specific proneuropeptides in juvenile and adult specimens. The authors use whole-mount stainings, tissue sections and tissue 'peals' to provide whole-body overviews and more detailed regional views at cellular resolution of nervous system organisation. This manuscript is the to date most comprehensive compendium of molecular nervous system characterization in this enigmatic animal group. The authors combine immunohistochemistry, whole mount in situ hybridization, multi-color HCR, dye-filling and transient transgenesis. 

The authors also extensively reviewed the available data about overall nervous system architecture in hemichordates and placed their findings in a broad context.

The manuscript is of high quality and will provide a valuable resource for researchers working on nervous system evolution and hemichordate biology. 

Minor comments:

The text is very long and the authors may want to consider shortening it or reorganising it. Some sections that would belong to the introduction or discussion are in the main text etc. See pdf for more detailed comments.

Figure 3 is very dense, the authors could consider breaking it up into two figures

In the pdf Figure 8 seems to have lost most of its labels, please check

In Figure 8 the figure legend is incomplete, lacking the description of the lower two panel-rows

Please see the attached pdf for more detailed comments on the manucript

"Our data provides some insights into this question, but also raises many others." - many questions

---

## [Decision Letter · Decision Letter 2]

11 Jul 2023

Dear Dr Andrade Lopez,

Thank you for the submission of your revised Methods and Resources article, "Molecular characterization of nervous system organization in the hemichordate acorn worm Saccoglossus kowalevskii" for publication in PLOS Biology. On behalf of my colleagues and the Academic Editor, Yi-Hsien Su, I'm pleased to say that we can in principle accept your manuscript for publication, provided you address any remaining formatting and reporting issues. These will be detailed in an email you should receive within 2-3 business days from our colleagues in the journal operations team; no action is required from you until then. Please note that we will not be able to formally accept your manuscript and schedule it for publication until you have completed any requested changes.

Sincerely, 

Roli Roberts

Senior Editor

PLOS Biology

rroberts@plos.org

REVIEWERS' COMMENTS:

Reviewer #1:

[identifies himself as Gerhard Schlosser]

The revision addresses all my previous comments (apart from being still deliberately vague about the proposed homologies to pituitary/hypothalamus).

Reviewer #3:

The authors have extensively edited the manuscript based on the reviewers' comments and have addressed all my comments. An important paper that should now be published without delay.